# MUL1 acts in parallel to the PINK1/parkin pathway in regulating mitofusin and compensates for loss of PINK1/parkin

Jina Yun[1,2], Rajat Puri[3†], Huan Yang[1†], Michael A Lizzio[1‡], Chunlai Wu[4], Zu-Hang Sheng[3], Ming Guo[1,2,5]*

[1]Department of Neurology, University of California, Los Angeles, Los Angeles, United States; [2]Molecular and Medical Pharmacology, University of California, Los Angeles, Los Angeles, United States; [3]Synaptic Functions Section, National Institute of Neurological Disorders and Stroke, National Institutes of Health, Bethesda, United States; [4]Neuroscience Center of Excellence, Louisiana State University Health Sciences Center, New Orleans, United States; [5]Brain Research Institute, The David Geffen School of Medicine, University of California, Los Angeles, Los Angeles, United States

*For correspondence: mingfly@ucla.edu

†These authors contributed equally to this work

Present address: ‡Department of Cell Biology and Molecular Genetics, University of Maryland, College Park, United States

Competing interests: The authors declare that no competing interests exist.

**Abstract** Parkinson's disease (PD) genes *PINK1* and *parkin* act in a common pathway that regulates mitochondrial integrity and quality. Identifying new suppressors of the pathway is important for finding new therapeutic strategies. In this study, we show that *MUL1* suppresses *PINK1* or *parkin* mutant phenotypes in *Drosophila*. The suppression is achieved through the ubiquitin-dependent degradation of Mitofusin, which itself causes *PINK1/parkin* mutant-like toxicity when overexpressed. We further show that removing *MUL1* in *PINK1* or *parkin* loss-of-function mutant aggravates phenotypes caused by loss of either gene alone, leading to lethality in flies and degeneration in mouse cortical neurons. Together, these observations show that *MUL1* acts in parallel to the *PINK1/parkin* pathway on a shared target *mitofusin* to maintain mitochondrial integrity. The *MUL1* pathway compensates for loss of *PINK1/parkin* in both *Drosophila* and mammals and is a promising therapeutic target for PD.

## Introduction

Parkinson's disease (PD) is the second most common neurodegenerative disorder and there is no cure for this progressive illness (*Guo, 2012*). Mutations in PINK1, a mitochondria-localized serine–threonine kinase, and Parkin, an E3 ubiquitin ligase, lead to autosomal recessive forms of the disease (*Kitada et al., 1998*; *Valente et al., 2004*). Genetic studies in *Drosophila* first demonstrated that *PINK1* and *parkin* act in the same genetic pathway, with *PINK1* positively regulating *parkin*, to regulate mitochondrial integrity and function (*Clark et al., 2006*; *Park et al., 2006*; *Yang et al., 2006*). Mitochondrial morphology is maintained by a balance between two opposing actions, mitochondrial fusion that is promoted by *mitofusin (mfn)* and mitochondrial fission that is controlled by *Dynamin-related protein 1 (Drp1)* (*Chan, 2012*; *Nunnari and Suomalainen, 2012*). Genetic studies in *Drosophila* have shown that downregulation of *mfn* or overexpression of *drp1* suppresses multiple phenotypes associated with lack of *PINK1* or *parkin*, including defects in mitochondrial integrity, cell death, tissue health, and flight ability (*Deng et al., 2008*; *Poole et al., 2008*; *Yang et al., 2008*). Parkin ubiquitinates Mfn and promotes Mfn degradation (*Poole et al., 2010*; *Ziviani et al., 2010*). However, it is not clear if increased *mfn* or decreased *drp1* levels are sufficient to cause the phenotypes observed in *PINK1* or *parkin* mutants.

**eLife digest** Parkinson's disease is the second most common neurodegenerative disorder. Symptoms include tremors, rigidity, and slowness, as well as dementia and depression. While most cases of Parkinson's disease have no known genetic cause, mutations in either of two genes—*PINK1* or *parkin*—are known to lead to the disease.

PINK1 and parkin belong to a single pathway that regulates the structure and function of mitochondria, the organelles that generate energy inside cells. Identifying inhibitors of this pathway is critically important for development of future therapies. In addition, previous studies showed that mice with mutations in *PINK1* or *parkin*, as opposed to those in humans and flies, display subtle signs of Parkinson's disease: the fact that these are weak suggests that other unknown proteins or cellular pathways might compensate for loss of the genes.

Yun et al. have now identified one such protein by showing that an increase in the level of a Protein called MUL1 counteracts the deleterious effects due to the loss of PINK1 or parkin in fruit flies. MUL1 is a mitochondrial protein that regulates another protein called mitofusin; the role of mitofusin is to promote the fusion of mitochondria. Conversely, removing MUL1 from PINK1 or parkin mutant worsens the symptoms because MUL1 is no longer present to compensate for the defects.

Yun et al. also show that MUL1 operates through a pathway that is independent of PINK1/parkin. Moreover, this pathway is found in both flies and mouse neurons, which suggest that it has been conserved during evolution. The work of Yun et al. also suggests that MUL1 as a potential therapeutic target for Parkinson's disease.

In addition to mitochondrial dynamics, the *PINK1/Parkin* pathway promotes mitophagy, selective autophagic degradation of defective mitochondria in mammalian cells. Accumulation of mitochondrial damage can result in loss of mitochondrial membrane potential. This leads to recruitment of Parkin to the depolarized mitochondria, ultimately resulting in autophagic degradation of these mitochondria (*Narendra et al., 2008*; *Ding et al., 2010*; *Gegg et al., 2010*; *Geisler et al., 2010*; *Matsuda et al., 2010*; *Narendra et al., 2010*; *Okatsu et al., 2010*; *Tanaka et al., 2010*; *Vives-Bauza et al., 2010*; *Chan et al., 2011*). Parkin-mediated mitophagy also occurs in mouse cortical neurons and heart muscle (*Cai et al., 2012*; *Chen and Dorn, 2013*). An important step during this process is Parkin-dependent ubiquitination of Mfn and other substrates, followed by their proteasome-dependent degradation (*Tanaka et al., 2010*; *Chan et al., 2011*). Relevant to PD, *PINK1* and *parkin* mutant fibroblasts from PD patients also show deregulation of mitochondrial dynamics and modest defects in the clearance of mitochondria (*Rakovic et al., 2011*, *2013*).

An important puzzle in the field of PD research is why mice lacking *PINK1* or *parkin* bear only subtle phenotypes related to dopaminergic neuronal degeneration or mitochondrial morphology change (*Palacino et al., 2004*; *Perez and Palmiter, 2005*; *Perez et al., 2005*; *Kitada et al., 2007*; *Frank-Cannon et al., 2008*; *Gautier et al., 2008*; *Gispert et al., 2009*; *Kitada et al., 2009*; *Akundi et al., 2011*). This raises the possibility that other mechanisms may compensate for loss of *PINK1* or *parkin*. Indeed, when *parkin* is knocked down in adult dopaminergic neurons rather than during development, more striking neuronal degeneration is observed (*Dawson et al., 2010*; *Shin et al., 2011*; *Lee et al., 2012*). However, the molecular mechanisms by which loss of *PINK1/parkin* function can be compensated are not known.

Mitochondrial ubiquitin ligase 1 (MUL1), also known as mitochondrial-anchored protein ligase (MAPL) (*Neuspiel, 2008*), mitochondrial ubiquitin ligase activator of NF-kB (MULAN) (*Li et al., 2008*), or growth inhibition and death E3 ligase (GIDE) (*Zhang et al., 2008*), was identified as an E3 protein ligase by three independent groups. Work in mammalian systems shows that MUL1 has small ubiquitin-like modifier (SUMO) ligase activity, stabilizing Drp1 (*Harder et al., 2004*; *Braschi et al., 2009*), or ubiquitin ligase activity, degrading Mfn (*Lokireddy et al., 2012*). As expected from a protein with these proposed biochemical activities, *MUL1* expression in mammalian cells results in smaller and more fragmented mitochondria (*Li et al., 2008*; *Neuspiel, 2008*). However, the consequences of loss of *MUL1* in vivo have not been reported in any organism.

In this study, we show that overexpression of *mfn* is sufficient to recapitulate many *PINK1/parkin* mutant phenotypes, underlining the central importance deregulation of this protein has for PD

pathogenesis. Expression of wild-type MUL1, but not a ligase-dead version, suppresses *PINK1* or *parkin* mutant phenotypes, and those due to *mfn* overexpression in *Drosophila*. Conversely, removing *MUL1* in *PINK1* or *parkin* null mutants results in enhanced phenotypes as compared with the single mutants, suggesting that *MUL1* acts in parallel to the *PINK1/parkin* pathway. MUL1 physically binds to Mfn and promotes its ubiquitin-dependent degradation. MUL1, but not a ligase-dead version, also regulates Mfn levels and mitochondrial morphology in human cells. Experiments in *Drosophila* and mammalian systems suggest that *MUL1* regulates *mfn* through a pathway parallel to that of *PINK1/parkin* pathway. Finally, knockdown of *MUL1* from *parkin* knockout mouse cortical neurons augments mitochondrial damage and induces neurodegeneration-like phenotypes than does removing either gene alone. Together, these results suggest that *MUL1* plays an important compensatory function in organisms or cells lacking *PINK1/parkin*.

## Results

### Overexpression of *MUL1*, but not a ligase-dead form, suppresses *PINK1* and *parkin* mutant phenotypes in dopaminergic neurons and muscle

We identified *MUL1* as a novel suppressor of *PINK1/parkin* mutant phenotypes. Human MUL1 contains two transmembrane (TM) domains and a highly conserved C-terminal ring finger (RNF) domain. Topological studies suggest that the two TM domains anchor the protein to the mitochondrial outer membrane, with the RNF domain facing the cytosol (*Li et al., 2008*). *Drosophila MUL1* (CG1134) encodes a protein with a similar domain structure, and 52% amino acid similarity to human MUL1 (*Figure 1A,B*).

We overexpressed *MUL1* in various tissues using the UAS/GAL4 system (*Brand and Perrimon, 1993*). *Drosophila* contains clusters of dopaminergic (DA) neurons in the adult brain. In wild-type DA neurons, mitochondria are dispersed in the cytosol (*Figure 1C*). In contrast, *PINK1* mutant DA neurons show abnormally clumped mitochondria (*Figure 1C'*, arrowheads) (*Park et al., 2006*), which can be suppressed by overexpression of *MUL1* (*Figure 1C''*).

We further characterized *MUL1*'s effects on *PINK1/parkin* mutants in thoracic indirect flight muscle (IFM), which consists of well-organized muscle fibers, in which mitochondria fill spaces between myofibrils. *PINK1* null mutant flies have severe defects in mitochondrial morphology, including an overall reduction in mitochondria-targeted GFP (mitoGFP) signal and the presence of large mitoGFP clumps (*Figure 1D–D',E–E'*). *PINK1* mutant muscle also shows extensive TUNEL-positive cell death (*Figure1E–E'*), muscle vacuolation, and degeneration (*Figure 1F–F'*). In addition, when examined under the electron microscopy (EM) level, many mitochondria are swollen with broken cristae (*Figure 1Ga–Ga',Gb–Gb'*). At the level of the whole organism, *PINK1* mutants show a thoracic indentation due to IFM degeneration (*Figure 1H*). Strikingly, *MUL1* overexpression almost completely rescues all of the above *PINK1* mutant phenotypes (*Figure 1D''–F'',Ga''–Gb'',H*).

To determine if the E3 ligase activity of MUL1 is required for suppression of *PINK1* mutant phenotypes, we generated a ligase-dead form of *Drosophila* MUL1 (MUL1 LD) in which histidine 307, a highly conserved residue within the RNF domain, was mutated to alanine (*Figure 1A,B*). This mutation has been shown to abolish ligase activity of mammalian MUL1 (*Zhang et al., 2008*); in vitro ubiquitination assays confirm that *Drosophila* MUL1 LD lacks ligase activity (*Figure 1—figure supplement 1*). The expression levels of *MUL1* and *MUL1* LD in muscles are comparable (*Figure 1I*), and no mitochondrial clumps or muscle cell death are observed when *MUL1* or *MUL1* LD is overexpressed in wild-type animals (*Figure 1J,J',K,K'*). Expression of *MUL1* LD does not suppress *PINK1* mutant phenotypes (*Figure 1H,J'',K''*). Overexpression of *MUL1* (*Figure 1L',M'*), but not *MUL1* LD (*Figure 1L'',M''*), also suppressed *parkin* null mutant phenotypes. Thus, *MUL1* is a robust suppressor of *PINK1/parkin* mutants and this requires MUL1's ligase activity.

### *MUL1* regulates mitochondrial morphology in *Drosophila*

As an E3 ligase anchored onto the mitochondrial outer membrane, MUL1 has been shown to have multiple substrates including Drp1 and Mfn (*Braschi et al., 2009*; *Lokireddy et al., 2012*). However, the consequences of loss of *MUL1* have not been reported in any organism. The P element, *MUL1^EY12156* (*MUL1^EY*), inserted at 20 bp upstream of the *MUL1* start codon (*Figure 2A*), is a partial loss-of-function allele with reduced mRNA expression (*Figure 2B–C*). We performed imprecise excision of this P element and generated a large deletion allele, *MUL1^A6*. *MUL1^A6*, hereafter called the *MUL1* mutant,

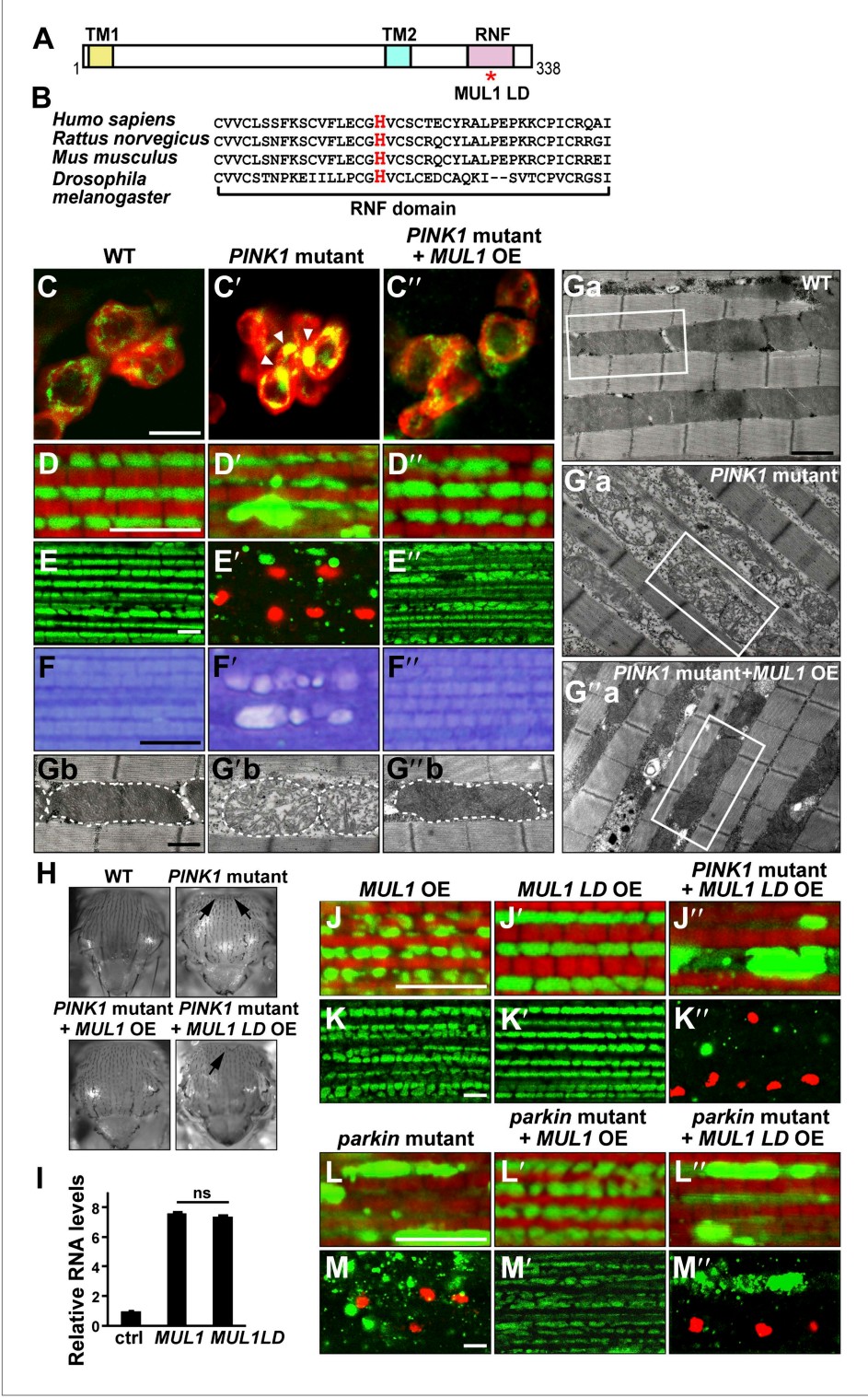

**Figure 1**. Overexpression of *MUL1*, but not *MUL1 LD*, suppresses *PINK1/parkin* mutant phenotypes. (**A**) Protein domain organization of Drosophila MUL1. TM1, TM2, and RNF represent transmembrane domains 1 and 2, and the RING Finger domain, respectively. The position of the mutation in the ligase dead (LD) version of MUL1 is marked with a red asterisk. (**B**) Sequence alignment of MUL1 in various species in the highly conserved RNF domain. A highly conserved histidine residue (marked as red) was mutated to alanine in MUL1 LD, ablating ligase activity. (**C–C''**) Dopaminergic neurons stained with an anti-TH antibody in red and mitochondria labeled with mitoGFP in

*Figure 1. Continued on next page*

*Figure 1. Continued*

green. Neurons in the PPL1 cluster are shown. While mitochondria in wild-type dopaminergic neurons are dispersed (**C**), mitochondria in *PINK1* mutant dopaminergic neurons are clumped (**C'**, white arrow heads). This phenotype is suppressed by *MUL1* overexpression driven by TH-Gal4 (**C"**). Scale bars: 10 µm. (**D–E""** and **J–M""**) Confocal images of the IFM from thoraces double labeled with mitoGFP and phalloidin (red) (**D–D"**, **J–J"**, **L–L"**), or double labeled with mitoGFP and TUNEL (red) with lower magnification (**E–E"**, **K–K"**, **M–M"**). Scale bars: 5 µm. *MUL1* overexpression is driven by Mef2-Gal4. In wild-type (**D**), mitochondria have a regular size and shape, and are localized in between myofibrils. In *PINK1* mutants (**D'**), mitochondrial size becomes irregular, and the GFP signal is reduced. Large mitochondrial clumps also appear. *PINK1* mutant muscle is TUNEL-positive (**E'**). (**F–F"**) Touidine blue staining of muscle. Compared with the wild-type (**F**), *PINK1* mutant muscle shows vacuolation indicating muscle degeneration (**F'**). These *PINK1* mutant phenotypes (**D'**, **E'**, **F'**) are almost completely suppressed by *MUL1* overexpression (**D"**, **E"**, **F"**). (**Ga–Ga"**, **Gb–Gb"**) EM images of mitochondria in muscle. (**Gb–Gb"**) Single mitochondrion (outlined with dashed lines) from white boxes in **Ga–Ga"**. Scale bars: 1 µm (**Ga–G"a**) 0.5 µm (**Gb–G"b**). In wild-type (**Ga** and **Gb**), mitochondria have compact and organized cristae whereas mitochondria from *PINK1* mutants (**Ga'**, **Gb'**) are swollen with fragmented cristae, and this is rescued by *MUL1* overexpression (**Ga"**, **Gb"**). (**H**) Images of thoraces. Arrows point to thoracic indentations due to muscle degeneration. Compared with WT, *PINK1* mutants have thoracic indentation due to muscle degeneration. *MUL1* overexpression, but not *MUL1 LD* overexpression, suppresses *PINK1* mutant thoracic indentation. (**I**) qPCR analysis shows that *MUL1* and *MUL1 LD* mRNA are expressed at similar levels in muscles. The data are shown as the mean ± SEM from three experiments (RNA from ten 5-day-old fly thoraces for each genotype). The statistical analysis was done using One-way ANOVA with Tukey' multiple comparisons test. ns: not statistically significant. *MUL1 LD* overexpression in the *PINK1* mutant background does not suppress the formation of mitochondrial clumps (**J"**) or TUNEL-positivity (**K"**). (**L–M"**) Overexpression of *MUL1*, but not *MUL1 LD*, suppresses *parkin* mutant phenotypes.

The following figure supplements are available for figure 1:

**Figure supplement 1**. MUL1, but not its ligase-dead version (MUL1 LD), is able to self-ubiquitinate in vitro.

---

produces no detectable transcript (*Figure 2B–C*), and therefore is a null allele. Flies homozygous for *MUL1^{A6}* are viable. We also generated two independent RNAi constructs that target two different locations in the *MUL1* coding region. Flies expressing these constructs (*MUL1* RNAi lines) show the same phenotypes (see below) and reverse the suppression of *PINK1* mutant phenotypes observed upon *MUL1* overexpression (*Figure 2D*).

We examined phenotypes due to loss-of-function and overexpression of *MUL1* in the IFM, which is a cellular syncytium, and in salivary glands, in which a number of individual cells contain an extensive tubular mitochondrial reticulum. Cells from *MUL1* null mutant flies, or flies in which *MUL1* RNAi is expressed, have mildly elongated mitochondria, while those from flies overexpressing *MUL1* have small and fragmented mitochondria (*Figure 2E–G*). Thus, *Drosophila MUL1* has a mild pro-fission function, as with mammalian *MUL1* (*Braschi et al., 2009*).

## MUL1 binds to Mfn and negatively regulates its levels through ubiquitination

Next, we asked whether Drp1, Mfn or both serve as MUL1 targets. Previous work suggested that MUL1 positively regulates Drp1's pro-fission activity through sumoylation-dependent protein stabilization (*Harder et al., 2004*; *Braschi et al., 2009*). Surprisingly, overexpression of *MUL1* did not change Drp1 levels (*Figure 3A*). In contrast, overexpression of *MUL1* led to a reduction in Mfn levels (*Figure 3B*). Analysis of larval lysates from mutants showed that loss of *MUL1* results in an increase in Mfn levels, as does loss of *PINK1* or *parkin*, which serve as positive controls (*Figure 5—figure supplement 1*). Knockdown of *MUL1* in *Drosophila* S2 cells also resulted in an increase in Mfn levels (*Figure 3C*).

Next, we determined if MUL1 and Mfn interact physically, and if MUL1 regulates Mfn levels through ubiquitination. Overexpressed MUL1 co-immunoprecipitated with overexpressed Mfn from *Drosophila* S2 cell lysates. Parkin was used as a positive control and co-immunoprecipitated with Mfn as previously reported (*Poole et al., 2010*; *Ziviani et al., 2010*). These physical interactions are specific to MUL1 and Mfn rather than the tags utilized (*Figure 3—figure supplement 1*). To assay ubiquitination, Flag-tagged Mfn was expressed in S2 cells exposed to dsRNA targeting *MUL1* or *parkin* in the presence of the proteasome inhibitor MG132 (*Figure 3E*). In control cells, highly ubiquitinated Mfn was observed. When cells were treated with two different dsRNAs against *MUL1*, ubiquitinated Mfn levels were

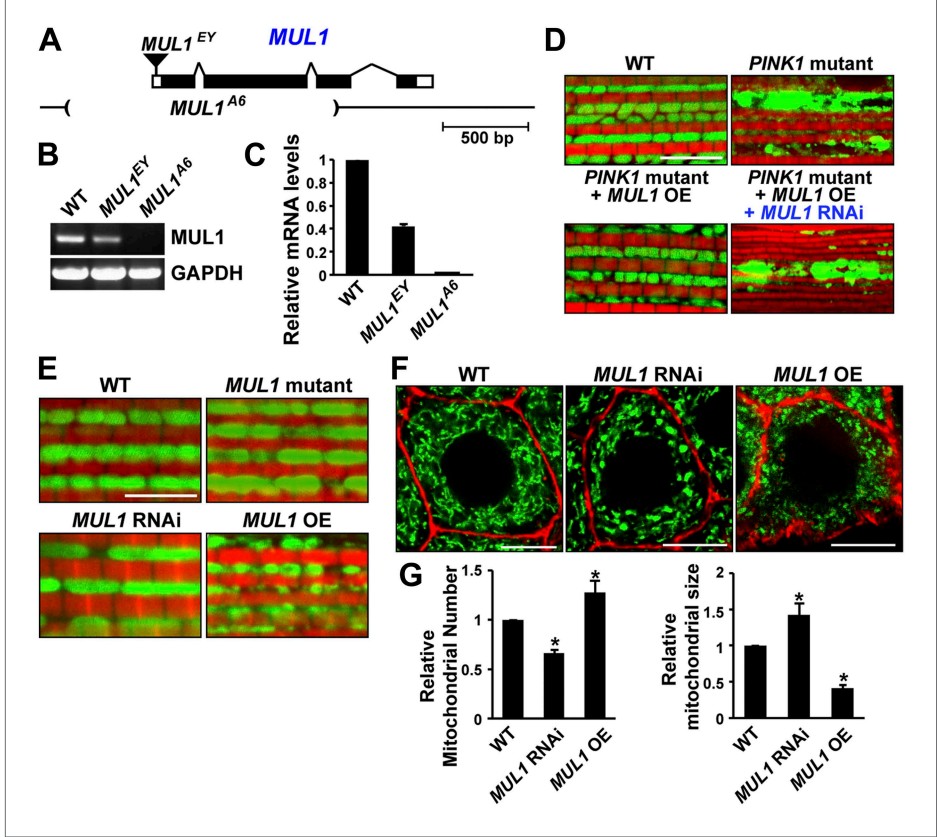

**Figure 2**. *MUL1* regulates mitochondrial morphology. (**A**) A schematic depicting the Drosophila *MUL1* genomic region (cytological location 64A4). *MUL1* coding and untranslated regions (dark and open rectangles, respectively) are depicted. The P element, *MUL1^{EY}*, inserted in the 5' UTR, is shown as an inverted triangle. The deleted region in the *MUL1^{A6}* allele is indicated by parentheses. (**B**) RT PCR shows that flies carrying the *MUL1^{EY}* allele have detectable but reduced levels of *MUL1* transcripts. However, no *MUL1* transcript is detected in flies homozygous for the *MUL1* deletion, *MUL1^{A6}*. (**C**) qPCR shows that *MUL1^{EY}* allele has approximately a 60% reduction of *MUL1* transcript compared to the wild-type (WT). No *MUL1* transcript is detected in flies homozygous for *MUL1^{A6}*. (**D**) *MUL1* RNAi line reverses the suppression of *PINK1* mutant mitochondrial phenotypes due to *MUL1* overexpression. (**E**) Muscle fibers stained with mitoGFP in green and actin in red. Compared with the WT, flies homozygous for the *MUL1* deletion or expressing *MUL1* RNAi show slightly elongated mitochondria. In contrast, when *MUL1* is overexpressed using the Mef2-Gal4 driver, mitochondria are significantly smaller. (**F**) Salivary glands, with cell boundaries labeled with rhodamine phalloidin in red, and mitoGFP in green. In WT, mitochondria are tubular and evenly distributed. In contrast, in cells expressing *MUL1* RNAi (driven by OK6-Gal4) mitochondria are fewer in number and found in clumps. In contrast, *MUL1* overexpression (also driven by OK6-Gal4) results in fragmented mitochondria and irregular cell boundaries. (**G**) Quantification of mitochondrial number and size in salivary glands (mean ± SEM, n > 6 larvae for each genotype). * Significantly different from wild-type, p<0.05 (One-way ANOVA with Tukey's multiple comparisons test).

dramatically reduced, similar to those observed in *parkin* RNAi-treated cells, which serve as a positive control. Finally, we observed that the increased Mfn levels seen in the *PINK1* mutant flies were reduced when *MUL1* was overexpressed (*Figure 3F*), strengthening our argument that MUL1 suppresses *PINK1* mutant phenotypes through reduction of Mfn levels. Together, these results suggest that MUL1 suppresses *PINK1/parkin* phenotypes by reducing Mfn levels through its ubiquitination-dependent degradation.

### *mfn* overexpression, but not loss of *drp1*, results in phenotypes similar to those of *PINK1* or *parkin* mutants; and these phenotypes are suppressed by *MUL1* overexpression

Previous studies showed that downregulation of *mfn* or overexpression of *drp1* could suppress *PINK1* and *parkin* mutant phenotypes in *Drosophila* (*Deng et al., 2008*; *Poole et al., 2008*; *Yang et al., 2008*).

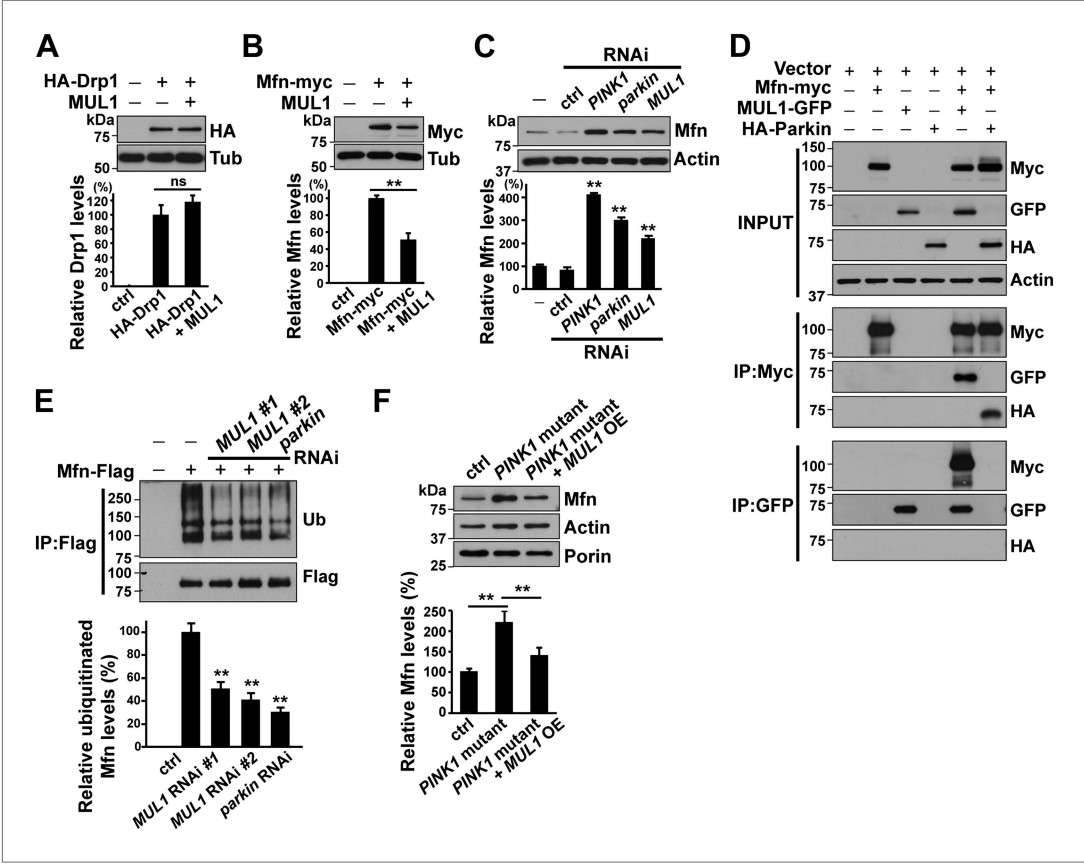

**Figure 3**. MUL1 physically binds to Mfn, and promotes ubiquitination-mediated Mfn degradation. (**A** and **B**) Western blots and quantifications of Drp1 and Mfn levels in vivo. Analysis of lysates from thoraces show that *MUL1* overexpression reduces Mfn levels (**B**) but not Drp1 levels (**A**). The data are shown as the mean ± SEM from three experiments (each experiment was done with lysate from 8 thoraces for each genotype). The statistical analysis was done using One-way ANOVA with Tukey's multiple comparisons test. ns: not statistically significant. ** Significantly different, p<0.01. (**C**) Western blots of Mfn levels in S2 cells either not treated or treated with control, *PINK1*, *parkin* or *MUL1* RNAi. Quantification of relative Mfn levels shows that there is an increase in Mfn levels in cells treated with RNAi to *PINK1*, *parkin*, or *MUL1* (mean ± SEM, ** Significantly different from cells not treated with RNAi, p<0.01, One-way ANOVA with Tukey's multiple comparisons test). (**D**) Co-immunoprecipitation using lysates from S2 cells transfected with the indicated constructs. The INPUT represents 2% of total lysate to monitor protein expression (top panel). MUL1-GFP is co-immunoprecipitated with Mfn-myc using both anti-GFP and anti-Myc antibodies. Mfn-myc also co-immunoprecipitates with HA-Parkin, which serves as a positive control. The interaction between Mfn-Myc and MUL1-GFP was specific, as confirmed by separate immunoprecipitation control experiments (***Figure 3—figure supplement 1***).
(**E**) Mfn ubiquitination levels in S2 cells. S2 cells are treated with dsRNA designed to silence various genes and transfected with Mfn-Flag. Immunoprecipitation was performed with anti-Flag antibody, and Western blots were probed with anti-Ubiquitin antibody and an anti-Flag antibody. Relative ubiquitination levels compared to control are shown below (mean ± SEM). ** Significantly different from control, p<0.01 (One-way ANOVA with Tukey's multiple comparisons test). In S2 cells, Mfn is highly ubiquitinated. RNAi of *MUL1* or *parkin* results in reduced levels of ubiquitnated Mfn. Two independent *MUL1* RNAs are utilized to knockdown *MUL1*, which yield the same results. (**F**) In *PINK1* mutant thoraces, where Mfn levels are increased, *MUL1* overexpression (driven by Mef2-Gal4) reduces the increased Mfn levels. Relative Mfn levels compared to control are shown below (mean ± SEM). ** Significantly different, p<0.01 (One-way ANOVA with Tukey's multiple comparisons test).

The following figure supplements are available for figure 3:

**Figure supplement 1**. MUL1 co-immunoprecipitates with Mfn in S2 cells.

Parkin has also been shown to bind and ubiquitinate Mfn, promoting Mfn degradation (*Gegg et al., 2010*; *Poole et al., 2010*; *Tanaka et al., 2010*; *Ziviani et al., 2010*; *Chan et al., 2011*; *Glauser et al., 2011*). While increased Mfn levels are observed in *PINK1* or *parkin* mutants (*Poole et al., 2010*; *Ziviani et al., 2010*), it is unclear if these increased *mfn* levels are sufficient to cause the phenotypes observed in *PINK1* or *parkin* mutants. It is also unclear if a decrease in the levels of *drp1*, which can result in increased mitochondrial size through loss of fission, results in a phenotypically equivalent effect.

To address these questions, we generated transgenic flies carrying UAS-*mfn* (also called *Marf*, CG3869) and obtained two *drp1* null alleles, *drp1¹* and *drp1²* (*Verstreken et al., 2005*). Overexpression of *mfn* under the control of the muscle-specific (*mef2*) GAL4 driver resulted in organismal lethality. To circumvent this lethality, we also generated a new Gal4 driver, IFM-GAL4, in which GAL4 expression is driven specifically in the IFM (*Figure 4F–J* for IFM-GAL1, vs *Figure 4A–E* for mef2-GAL4), using regulatory sequences from the *flightin* gene. Since the IFMs are not required for viability, knockdown of essential genes using IFM-GAL4 does not cause lethality in flies (data not shown).

Interestingly, overexpression of *mfn* in the IFM results in phenotypes (*Figure 4M,P,Sa,Sb*) similar to those of *PINK1* or *parkin* mutants; mitoGFP clumps, TUNEL-positive muscle cell death, and broken mitochondrial cristae when examined at the EM level (*Figure 1D',E',G'a,G'b*). In contrast, while loss of *drp1* results in an increase in mitochondrial size, no muscle cell death or degeneration is observed (*Figure 4V,Y*). Importantly, *MUL1* overexpression (*Figure 4T,W*), as with *parkin* overexpression (*Figure 4U,X*), suppressed the phenotypes associated with *mfn* overexpression. Together, these results show that overexpression of *mfn*, but not loss of *drp1*, leads to phenotypes similar to those due to lack of *PINK1* or *parkin*, suggesting a direct link between increased Mfn levels and pathology.

## *MUL1* acts in parallel to the *PINK1/Parkin* pathway

Our observations that *MUL1* overexpression suppresses *PINK1* or *parkin* mutant phenotypes, and that both Parkin and MUL1 promote Mfn degradation, suggest two possible scenarios of how *MUL1* and *PINK1/parkin* interact. *MUL1* may be a downstream target of the *PINK1/parkin* pathway and upstream of *mfn*. Alternatively, *MUL1* could function in a parallel pathway to *PINK1/Parkin*, but with action on a common target such as Mfn. Characterization of double null mutants provides an effective way of distinguishing these possibilities. If *MUL1* functions in the same pathway as *PINK1*, double null mutants of *PINK1* and *MUL1* would be expected to show the same phenotype as the single mutant alone, as is observed in the case of *PINK1 parkin* double mutants (*Clark et al., 2006*; *Park et al., 2006*). Conversely, if *MUL1* and *PINK1/parkin* act in parallel pathways, the phenotypes of double null mutants may be stronger than those of single mutants.

We generated *PINK1 MUL1* and *parkin MUL1* double mutants. Several lines of evidence show that double mutants have significantly enhanced phenotypes as compared to those of single mutants alone. First, *PINK1 MUL1* and *parkin MUL1* double null mutants show a high frequency of pupal lethality as compared with single mutants (data not shown), while double null mutants of *PINK1 parkin* have the same level of viability as single mutants (*Clark et al., 2006*; *Park et al., 2006*). Second, a thoracic indentation observed in *PINK1* or *parkin* null mutants is much more severe in *PINK1 MUL1* and *parkin MUL1* double null mutants. In contrast, *PINK1 parkin* double null mutants show the same degree of thoracic indentation as *PINK1* or *parkin* single mutants alone (*Figure 5A–G*). Third, at the cellular level, *PINK1 MUL1* and *parkin MUL1* double null mutants have highly elongated and interconnected mitochondria, as determined using anti-mitochondrial ATPase antibodies. These mitochondrial phenotypes are very different from those of *PINK1*, *parkin*, or *MUL1* mutants (*Figure 5I–O*). *PINK1 parkin* double null mutants show similar mitochondrial morphology phenotypes as *PINK1* or *parkin* single mutants alone (*Figure 5O* vs *Figure 5J,M*). Fourth, ATP levels in *parkin MUL1* double null mutants were further reduced compared to those of *parkin* or *MUL1* single null mutants (*Figure 5Q*). Fifth, the ability of *parkin* overexpression to rescue *PINK1* mutants is not dependent on *MUL1*, and *MUL1* overexpression can still suppress *PINK1* mutants in the absence of *parkin* (*Figure 5—figure supplement 1*). Sixth, knockdown of both *MUL1* and *parkin* in S2 cells further reduces Mfn ubiquitination below levels seen with knockdown of *MUL1* or *parkin* alone (*Figure 5R*). Seventh, *PINK1 MUL1* and *parkin MUL1* double null mutants have higher Mfn levels as compared to single null mutants of *MUL1*, *PINK1*, or *parkin* (*Figure 5—figure supplement 1*). Finally, knockdown of *mfn* in the background of *parkin MUL1* double mutants almost completely rescues the thoracic indentation and mitochondrial phenotypes of *parkin MUL1* double mutants (*Figure 5H,P*). These genetic observations, in combination with biochemical findings that MUL1 physically interacts with Mfn, and that loss of *MUL1* results in decreased

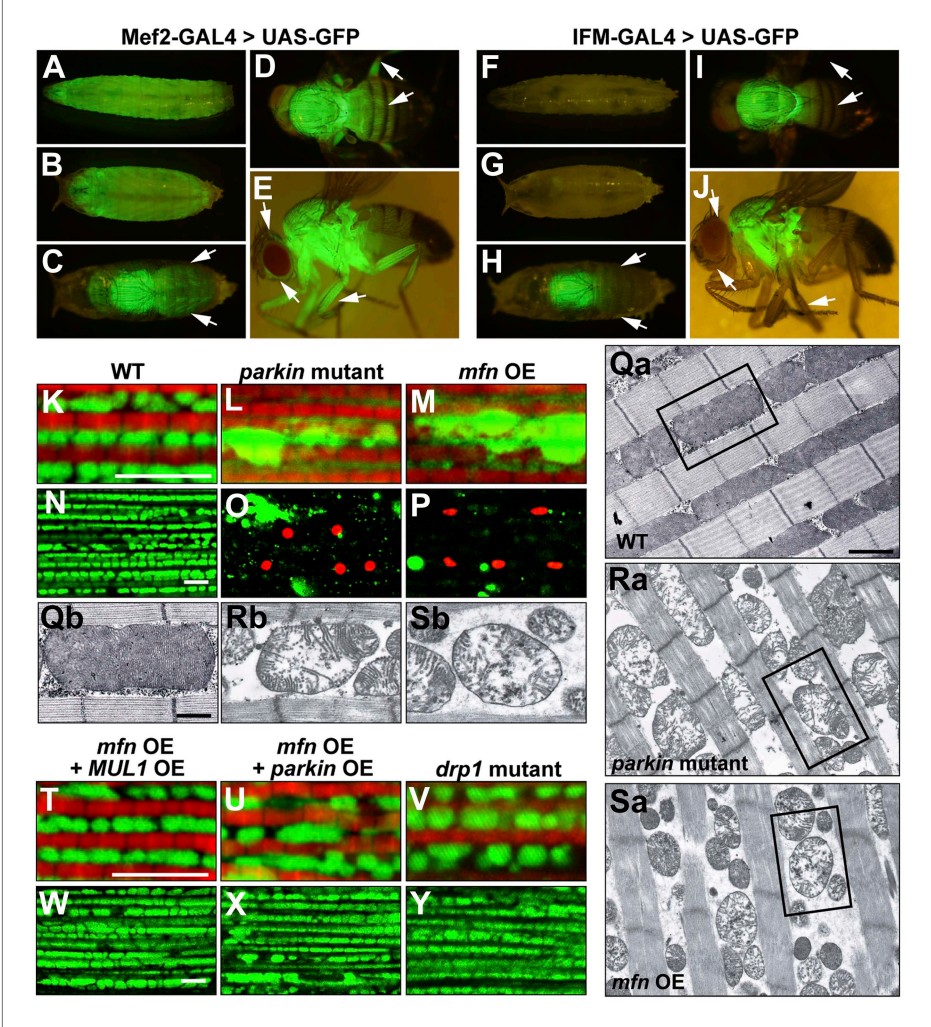

**Figure 4**. Generation and expression of the IFM-GAL driver; *mfn* overexpression, but not loss of *drp1*, induces *PINK1/parkin*-mutant like pathology. (**A–J**) Different developmental stages of flies expressing GFP under Mef2-Gal4 (**A–E**) or IFM-Gal4 (**F–J**). (**A**) Third instar larvae show GFP expression in whole body muscles. (**B**) At the early pupal stage, GFP is expressed in a similar pattern as in larvae. However, the GFP expression pattern become more specific at the late pupal stage (**C**), in which the strongest GFP signal is seen in the thorax, and a weaker signal is observed in the head and abdomen (arrows). (**D**) In an adult fly, dorsal view shows GFP signal in the thorax, upper abdomen and legs. (**E**) GFP is also expressed in adult head and legs, marked with arrows. (**F**) Flies expressing GFP under IFM-Gal4 show no GFP expression in third instar larvae, or in early pupae (**G**). (**H**) GFP is strongly expressed only in the thorax at the late pupal stage, but not in other areas (arrows). (**I**) In the adult fly, GFP signal is highly concentrated in the thorax. No GFP expression in abdomen and legs is observed, arrows. (**J**) In contrast to GFP expression under Mef2-Gal4, IFM-Gal4 does not express in adult head or legs, as indicated with arrows. (**K–P, T–Y**) Confocal images of muscle double labeled with mitoGFP (green) and phalloidine (red) (**K–M, T–V**), or those labeled with mitoGFP and TUNEL (red) at lower magnification (**N–P, W–Y**), respectively. (**Qa–Sb**) EM images of mitochondria in muscle. Single mitochondrion from the black-boxed area in **Qa, Ra, Sa** is shown in Qb, Rb, Sb. Scale bars: 1 µm (**Qa, Ra, Sa**) and 0.5 µm (**Qb, Rb, Sb**). Compared with wild-type (**K** and **N**), *parkin* null mutant (**L** and **O**) shows overall reduced levels of mitoGFP signal, large mitochondrial clumps, and muscle cell death. Similar phenotypes are observed with *mfn* overexpression (**M** and **P**), and these phenotypes are suppressed by *MUL1* overexpression (**T** and **W**). As a control, *parkin* overexpression also suppresses phenotypes due to *mfn* overexpression (**U** and **X**). Importantly, *drp1* null (*drp1¹/drp1²*) mutant muscle does not have any mitochondrial clumping or TUNEL-positivity seen in loss of *parkin* function or *mfn* overexpression (**V** and **Y**). *mfn* overexpression is driven by IFM-Gal4. Scale bars: 5 µm.

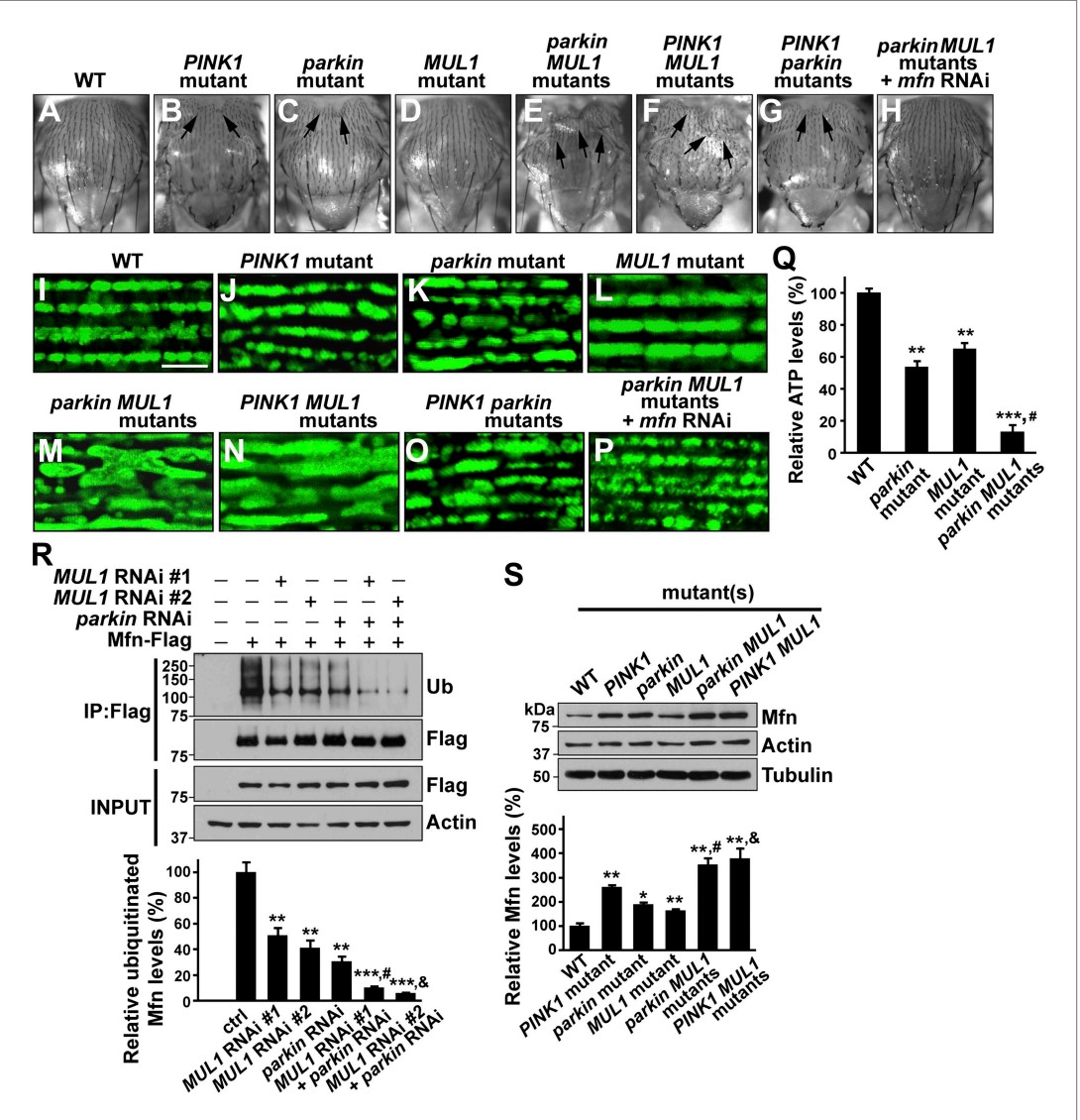

**Figure 5**. *MUL1* acts in parallel to the PINK1/parkin pathway. (**A–H**) Images of thoraces of various mutants. Arrows point to thoracic indentations due to muscle degeneration. *PINK1 MUL1* and *parkin MUL1* double mutants have more severe thoracic indentation compared to either mutant alone. Remarkably, the severe thoracic indentation phenotype in *parkin MUL1* double mutants is almost completely suppressed when *mfn* is also knocked down. (**I–P**) Mitochondria are labeled using an anti-ATP synthase antibody in the IFM. While *PINK1*, *parkin*, and *MUL1* mutant show slightly elongated mitochondrial morphology, *PINK1 MUL1* and *parkin MUL1* double mutants exhibit highly elongated and interconnected mitochondria. These phenotypes can be suppressed by *mfn* knockdown. Instead of using mitoGFP, we utilized anti-ATPase antibodies that allow better visualization of the enhancement phenotypes seen with double mutants. (**Q**) Relative ATP levels in whole flies of various mutants (mean ±SEM from three experiments, five 5-day-old flies for each genotype). ** and *** significantly different from wild-type, $p<0.01$ and $p<0.001$, respectably (One-way ANOVA with Tukey's multiple comparisons test). # Significantly different from *parkin* mutant and *MUL1* mutant, both $p<0.01$ (Two-way ANOVA with Tukey's multiple comparisons test). (**R**) In vivo ubiquitination assay of Mfn. S2 cells were treated with the indicated RNAi, transfected with Flag-Mfn, and treated with proteasome inhibitor MG132. Immunoprecipitations were performed using anti-Flag antibody, and western blots were probed with antibodies against anti-Ubiquitin antibody (P4D1) or anti-Flag antibody. Relative ubiquitination levels compared to control are shown in the lower panel (mean ± SEM). ** and *** Significantly different from control, $p<0.01$ and $p<0.001$, respectably (One-way ANOVA with Tukey's multiple comparisons test). # Significantly different from *MUL1* RNAi #1 and *parkin* RNAi, both $p<0.01$. & Significantly different from *MUL1* RNAi #2 and *parkin* RNAi, $p<0.001$ and $p<0.01$, respectively (Two-way ANOVA with Tukey's multiple comparisons test). (**S**) Western

*Figure 5. Continued on next page*

*Figure 5. Continued*

blot analysis of Mfn levels in vivo and quantification (mean ± SEM from three experiments, eight third instar larvae for each genotype). * and ** significantly different from wild-type, p<0.05 and p<0.01, respectably (One-way ANOVA with Tukey's multiple comparisons test). # Significantly different from *parkin* mutant and *MUL1* mutant, both p<0.01. & Significantly different from *PINK1* mutant and *MUL1* mutant, both p<0.01 (Two-way ANOVA with Tukey's multiple comparisons test).

The following figure supplements are available for figure 5:

**Figure supplement 1**. *MUL1* acts in a parallel pathway to the *PINK1/parkin* pathway.

---

ubiquitination of endogenous Mfn and increased Mfn levels, indicate that *MUL1* acts in parallel to the *PINK1/parkin* pathway to regulate a common target Mfn.

## The roles of *MUL1* in regulating mitochondrial morphology and mfn levels are conserved in human cells

Next, we asked if MUL1-mediated mitochondrial morphology and Mfn regulation is conserved in human cells. We expressed human *MUL1* and *MUL1 LD* in HeLa cells (*Figure 6F*). Cells expressing MUL1 or MUL1 LD are GFP-positive and marked with asterisks (*Figure 6A–D″*). Cells expressing GFP-MUL1 showed peri-nuclear mitochondrial clustering (*Figure 6A–B*, asterisks), and mitochondria appeared small and globular in shape as compared to those in untransfected, GFP-negative cells (*Figure 6B–B″*). MUL1 LD neither causes mitochondrial clustering nor alters mitochondrial morphology (*Figure 6C–D″*, asterisks).

Mammals have two Mfn proteins, Mfn1 and Mfn2, both able to promote mitochondrial fusion (*Chen et al., 2003*; *Eura et al., 2003*). We monitored the fate of Mfn1 and Mfn2 in control HeLa cells and HeLa cells stably expressing small hairpin RNA against *MUL1* (*MUL1* shRNA) (*Figure 6E,G*). When cells were exposed to the protein synthesis inhibitor cycloheximide (CHX), Mfn1 and Mfn2 levels gradually decreased in control cells, but were dramatically stabilized in cells with decreased levels of *MUL1* (*Figure 6E*). To confirm the above result, we generated *MUL1* knockout HeLa cells (*Figure 6H*), which contain a deletion including the start codon in the *MUL1* genomic region, using the CRISPR/Cas 9 system (*Cong et al., 2013*; *Jinek et al., 2013*; *Mali et al., 2013*). Two independent anti-MUL1 antibodies confirmed no MUL1 expression in *MUL1* knockout cells (*Figure 6J* and data not shown). Similar to what was observed for the *MUL1* shRNA, Mfn1 and Mfn2 levels were also dramatically stabilized in *MUL1* knockout cells (*Figure 6I*). Together, these results suggest that the role of MUL1 in regulating Mfn stability and mitochondrial morphology is conserved in human cells.

## *MUL1* does not affect Parkin-mediated mitophagy

The *PINK1/Parkin* pathway mediates mitophagy in HeLa cells (*Narendra et al., 2008*, *2010*), mouse cortical neurons, and heart muscle (*Cai et al., 2012*; *Chen and Dorn, 2013*). When cells are treated with a mitochondrial uncoupler, mitochondria lose their membrane potential. This leads to recruitment of Parkin to the depolarized mitochondria, ultimately resulting in autophagic degradation of these mitochondria. Because of the genetic interactions observed between *MUL1* and *PINK1/parkin* in *Drosophila*, we asked if *MUL1* was able to modulate Parkin-mediated mitophagy.

We induced mitophagy by exposing HeLa cells to antimycin A, which inhibits electron transport and depolarizes the mitochondrial membrane. Wild-type, *MUL1* knockout, and *PINK1* knockout (as a control) HeLa cells were transfected with YFP-Parkin and treated with DMSO or antimycin A for indicated time. In wild-type, without antimycin A treatment, Parkin mainly localizes in the cytosol (*Figure 7A*). However, following antimycin A treatment for 3 hrs, most Parkin was translocated to the mitochondria (*Figure 7B,D*). After 24 hrs of antimycin A treatment, Parkin was found dispersed in the cytosol and mitochondria were no longer observed, indicating that mitophagy had occurred (*Figure 7C,E*). To ensure there was no delay in mitophagy, we also assayed cells that were treated with antimycin A for 6, 12, 16 and 20 hrs (*Figure 7F*, data not shown). In all cells, mitophagy was observed at least 16 hrs after antimycin A treatment. No significant differences were observed in the fraction of Parkin recruited to mitochondria, or the fraction of mitochondria that underwent mitophagy, at any of these time points, for wild-type and *MUL1* knockout cells (*Figure 7B–F*). Knockdown of *MUL1* using shRNA also had no effect on Parkin translocation or mitophagy (*Figure 7—figure supplement 1*).

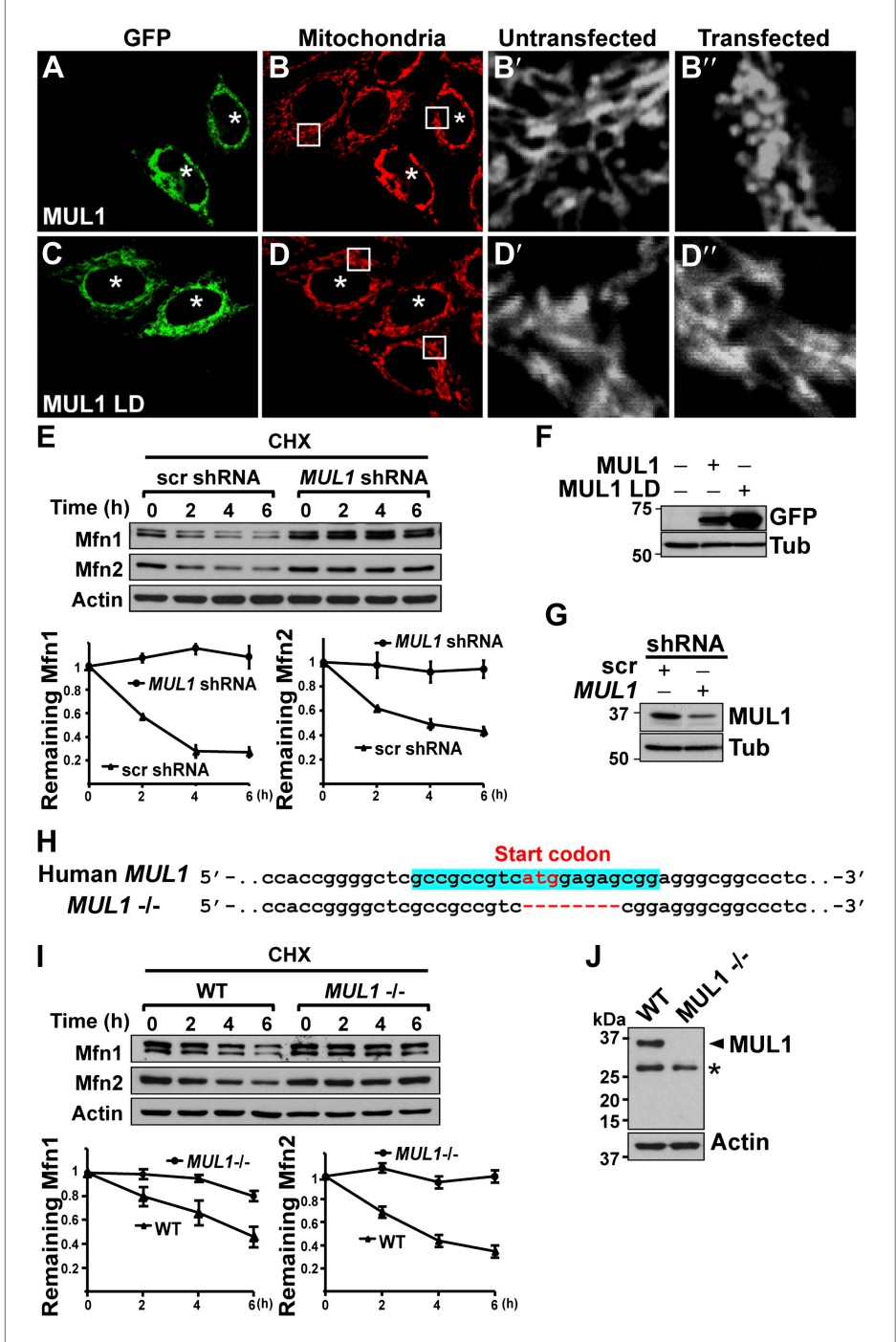

**Figure 6**. *MUL1*'s function in mitochondrial morphology and Mfn levels is conserved in human cells. (**A**–**D'''**) HeLa cells transfected with GFP-MUL1 (**A**–**B''**) or GFP-MUL1 LD (**C**–**D''**) are marked with asterisks, while cells not transfected serve as internal controls. Mitochondria are labeled with mitotracker in red (**B** and **D**). (**B'** and **B''**, **D'** and **D''**) Higher magnification images of mitochondria within white boxes in **B** and **D**. Cells expressing GFP-MUL1 have clustered mitochondria in the perinuclear region (**B**). Mitochondria are also small and fragmented (**B''**), as compared to cells not expressing GFP-MUL1 (**B'**). Importantly, GFP-MUL1 LD does not result in localization of mitochondria to the perinuclear region (**D**) or in mitochondrial fragmentation (**D'**). (**E**) Western blot analysis of Mfn1 and Mfn2 levels after CHX treatment. HeLa cells expressing scrambled shRNA or *MUL1* sh*MUL1* are treated with CHX. Mfn1 and 2 levels at each time point are normalized with Actin. The relative portion of remaining Mfn1 and 2 as compared to time point 0 was calculated and plotted (**E**). In cells expressing *MUL1* shRNA, Mfn1 and 2 levels after CHX

*Figure 6. Continued on next page*

*Figure 6. Continued*

treatment are more stable than those in cells expressing scrambled shRNA. (**F**) Expression of transfected GFP-MUL1 and GFP-MUL1 LD in HeLa cells, as detected using anti-GFP antibody. (**G**) Western blot analysis of endogenous MUL1 levels in HeLa cells stably expressing scrambled shRNA and *MUL1* shRNA. *MUL1* shRNA expressing cells have reduced levels of endogenous MUL1. (**H**) Human *MUL1* sequence and deletion in *MUL1* knockout (*MUL1−/−*) HeLa cells, generated using the CRISPR/Cas 9 system. Sequences targeting *MUL1* are highlighted in blue. Red letters indicate start codon. Red dashes represent deleted bases. Deleted eight base pairs include the start codon of MUL1. (**I**) Western blot analysis of Mfn1 and Mfn2 levels in wild-type and *MUL1−/−* HeLa cells treated with CHX for the indicated time. Remaining Mfn1 and Mfn2 levels at each time point were plotted below. (**J**) Western blot showing no MUL1 expression in *MUL1−/−* HeLa. Arrowhead points to MUL1 protein. Asterisk indicates a non-specific band.

As a control, *PINK1* knockout HeLa cells showed almost no Parkin localization to mitochondria and lack of mitophagy (*Figure 7B–F*). Similar results were obtained when cells were treated with carbonyl cyanide m-chlorophenylhydrazone (CCCP), which uncouples mitochondrial membrane potential (data not shown). Finally, *MUL1* overexpression also had no effect on Parkin translocation (*Figure 7G–H*) and did not block mitophagy (data not shown). Thus, neither loss of *MUL1* nor its overexpression altered Parkin translocation or mitophagy. These results are consistent with *MUL1* acting in parallel to *PINK1/parkin* in *Drosophila*, and suggest that *MUL1* regulates mitochondrial health through a distinct pathway.

## Loss of both *MUL1* and *parkin* aggravates mitochondrial damage and induces degeneration-like phenotypes in mouse cortical neurons

To further test the hypothesis that *MUL1* functions in parallel to the *PINK1/parkin* pathway in mammalian cells, we investigated the effects of depleting both *MUL1* and *parkin*. As HeLa cells do not express Parkin, we turned to cultured mature cortical neurons. GFP-MUL1 localizes to the mitochondria in cell bodies and axons of the primary cortical neurons (*Figure 8A*, *Figure 8—figure supplement 1*).

The proper maintenance of the mitochondrial inner membrane potential ($\Delta\psi_m$) depends on the physiological function of the mitochondrial respiratory chain, and is crucial for generating ATP. Dissipation of the membrane potential is a strong indication of unhealthy mitochondria, which can lead to severe mitochondrial dysfunction and subsequent cell death. The $\Delta\psi_m$ can be measured using a fluorescent dye tetramethyl rhodamine ethyl ester (TMRE). We used two independent MUL1 shRNAs to suppress endogenous MUL1 expression in cortical neurons (*Figure 8B*). Cortical neurons expressing CFP-mito, from either wild-type mice co-expressing two independent MUL1 shRNA (*Figure 8D–D'*), or *parkin* gene KO mice co-expressing a scrambled shRNA (*Figure 8E–E'*), showed no significant decrease in the intensity of TMRE fluorescence (*Figure 8C,C',G*). However, *parkin* KO neurons co-expressing *MUL1* shRNA showed a significant reduction of $\Delta\psi_m$ (*Figure 8F–F',G*), suggesting that proper *MUL1* expression in primary cortical neurons is required to compensate for loss of *parkin* in maintaining mitochondrial integrity.

Next, we asked if *MUL1* knockdown alters Mfn2 levels in mouse cortical neurons. Neurons from wild-type or *parkin* KO mice were transfected with either scrambled shRNA or *MUL1* shRNA, followed by co-staining with anti-Cytochrome C and anti-Mfn2 antibodies. Relative Mfn2 intensity in individual neurons was analyzed by calculating the ratio of Mfn2 to Cytochrome C (*Figure 8—figure supplement 2*). Compared to wild-type neurons transfected with scrambled shRNA, wild-type neurons with *MUL1* shRNA, or *parkin* KO neurons with either scrambled shRNA or *MUL1* shRNA had an increased intensity ratio of Mfn2 to Cytochrome C. This suggests that MUL1's role in regulating Mfn2 levels is also conserved in neurons.

Finally, we investigated if loss of either *MUL1* or *parkin,* or loss of both, has any impact on cultured primary cortical neuron morphology. *MUL1* knockdown in cells from wild-type mice resulted in a minor increase in dendritic retraction but no significant process fragmentation as compared with wild-type cells (*Figure 8H,L,M*). Cortical neurons from *parkin* KO mice showed slightly increased process fragmentation but no dendritic retraction (*Figure 8J,L,M*), as compared with wild-type cells. In contrast, *MUL1* knockdown in *parkin* KO neurons resulted in a dramatic increase in the number of neurons with dendritic and axonal fragmentation and retraction (*Figure 8K–K″,L–M*; process fragmentation: total number of neurons examined: n > 115 and n > 127 each genotype for fragmentation and dendritic

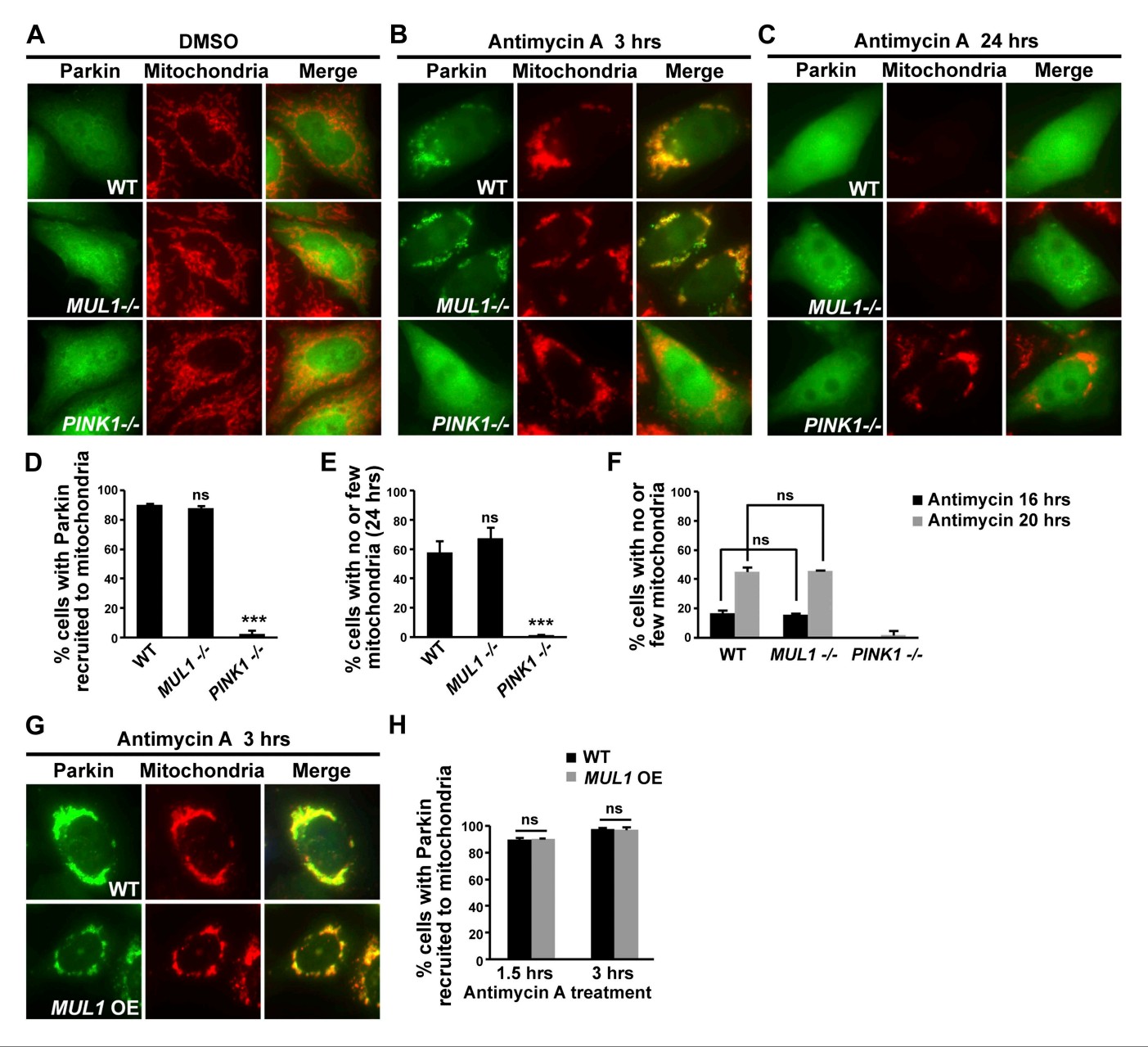

**Figure 7**. Neither *MUL1* knockout nor overexpression affects Parkin-mediated mitophagy. (**A**–**C**) HeLa cells (control, *MUL1* knockout or *PINK1* knockout) were transfected with YFP-Parkin, treated with either DMSO or antimycin A, and immunostained with an anti-Tom20 antibody which labels mitochondria. (**A**) HeLa cells treated with DMSO as a control. (**B**) Following treatment with antimycin A for 3 hrs, Parkin is recruited to mitochondria, as shown by co-localization of Parkin and the mitochondrial marker. In *MUL1* null cells, Parkin recruitment to mitochondria is not affected, whereas in *PINK1* null cells (positive control), Parkin recruitment to mitochondria is abolished. (**C**) After 24 hrs of antimycin A treatment, Parkin returns to the cytosol and the mitochondrial signal disappears. In *MUL1* null cells, mitochondrial disappearance occurs similarly as with WT, whereas in *PINK1* null cells (positive control), mitochondria are not eliminated. (**D**–**E**) Quantification of cells with Parkin recruited to mitochondria after 3 hrs of antimycin A treatment (**D**) and with few or no mitochondria after 24 hr of antimycin A treatment (**E**) and after 16 and 20 hrs of antimycin A treatment (**F**). The data are shown as the mean ± SEM from three experiments (n ≥ 100 for each genotype). *** Significantly different from wild-type, $p < 0.001$. ns: not statistically significant (One-way ANOVA with Tukey's multiple comparisons test). While Parkin translocation and mitochondrial disappearance are significantly blocked in *PINK1* knockout cells, there is no significant difference between HeLa cells and *MUL1* knockout cells in these processes. (**G**) HeLa cells stably expressing YFP-Parkin and mito-RFP are transfected with Flag-MUL1, treated with DMSO or antimycin A, and immunostained with anti-Flag antibody. 3-hour antimycin A treatment causes Parkin localization to mitochondria in cells with or without *MUL1* expression. (**H**) Quantification of cells with Parkin recruited to mitochondria after 1.5 or 3 hrs Antimycin A treatment. Both 1.5 and 3 hrs of antimycin A treatments results in similar levels of Parkin
*Figure 7. Continued on next page*

*Figure 7. Continued*

recruitment to mitochondria. The data are shown as the mean ± SEM from three experiments (n ≥ 100 for each genotype). ns: not statistically significant (One-way ANOVA with Tukey's multiple comparisons test).

The following figure supplements are available for figure 7:

**Figure supplement 1**. *MUL1* knockdown does not affect Parkin-mediated mitophagy.

retraction analysis, respectively), indicative of early neurodegeneration. The observed phenotypes were confirmed using a second *MUL1* shRNA in *parkin* KO neurons (data not shown). These observations suggest that *MUL1* acts in parallel to the *PINK1/parkin* pathway to ensure mitochondrial integrity and function, thus maintaining neuronal health in primary cortical neurons.

## Discussion

In summary, we identified *MUL1* as a robust suppressor of *PINK1/parkin* mutant phenotypes in *Drosophila*. *MUL1* overexpression, but not expression of a ligase-dead version, strongly suppresses *PINK1* and *parkin* mutant phenotypes. The mechanism of this suppression is unique in that *MUL1* does not act on *PINK1* or *parkin*, nor does it function as a downstream target. Rather, *MUL1* acts by suppressing *mfn* in parallel to the *PINK1/parkin* pathway (**Figure 9B**). *mfn* is crucial for actions downstream of *PINK1/parkin* to maintain mitochondrial function and tissue health (**Figure 9B**), as overexpression of *mfn* leads to pathology similar to lack of *PINK1/parkin* function. We hypothesize that the increase in the Mfn level needs to reach a threshold, such as that observed in the *PINK1/parkin* mutant backgrounds, but not in the *MUL1* null background, in order for overt muscle cell degeneration to occur (**Figure 9B**). Biochemically, MUL1 binds to Mfn, and loss of *MUL1* results in decreased ubiquitination of Mfn and increased Mfn levels. These observations suggest that MUL1 may directly ubiquitinate Mfn, leading to its degradation (**Figure 9A**). Alternatively, MUL1 may act via an intermediary that promotes Mfn ubiquitination and degradation. In *Drosophila*, overexpression of *MUL1* almost completely suppresses all aspects of the *PINK1/parkin* null phenotypes. Thus, treatments that manipulate *MUL1* expression or activity may have potential as therapeutics strategies.

In addition to showing that overexpression of *MUL1* compensates for lack of *PINK1/parkin* by downregulating Mfn levels, we have identified an evolutionarily conserved pathway and provide compelling evidence showing that endogenous levels of *MUL1* normally compensates for lack of *PINK1* or *parkin* in *Drosophila* and in mammals. Removal of *MUL1* in the *PINK1* or *parkin* null background significantly aggravates the phenotypes due to lack of *PINK1 or parkin* alone. Flies lacking *MUL1*, *PINK1* or *parkin* are viable, but *PINK1/MUL1* or *parkin/MUL1* double mutants manifest increased lethality with much more severe muscle degeneration, reduced ATP levels, defective mitochondrial morphology and increased Mfn levels. In addition, while *parkin* KO mature mouse cortical neurons or *MUL1* knockdown neurons show only mild neuronal phenotypes, neurons with *parkin* KO and *MUL1* KD show significantly diminished mitochondrial membrane potential, indicating mitochondrial dysfunction. They also show neurodegeneration-like phenotypes including axonal and dendritic fragmentation, and reduced mitochondrial distribution along processes. Finally, human HeLa cells, which have little or no endogenous Parkin, show a dramatic stabilization of Mfn when *MUL1* is eliminated.

Our findings may help to address an important puzzle in the field of PD research: why do *PINK1* or *parkin* knockout mice, or even *parkin/DJ-1/PINK1* triple knockout mice, bear only subtle phenotypes related to dopaminergic neuronal degeneration or mitochondrial morphology changes (***Palacino et al., 2004***; ***Perez and Palmiter, 2005***; ***Perez et al., 2005***; ***Kitada et al., 2007***; ***Frank-Cannon et al., 2008***; ***Gautier et al., 2008***; ***Gispert et al., 2009***; ***Kitada et al., 2009***; ***Akundi et al., 2011***). Our studies provide genetic and cellular clues that suggest compensation by *MUL1* may contribute to the subtle phenotypes in *PINK1* or *parkin* mutant mice. It will be interesting to determine whether *PINK1/MUL1* or *parkin/MUL1* double knockout mice show more severe PD-related pathology. Regarding PD therapies, optimizing the function of *MUL1* is likely to be beneficial for *PINK1/PARKIN* patients; upregulating *MUL1* may rescue the pathology due to lack of *PINK1* or *PARKIN*. In contrast, downregulating *MUL1* and/or mutations in *MUL1* may lead to disruption of this compensatory pathway in maintaining mitochondrial integrity and function, and result in accelerated disease progression.

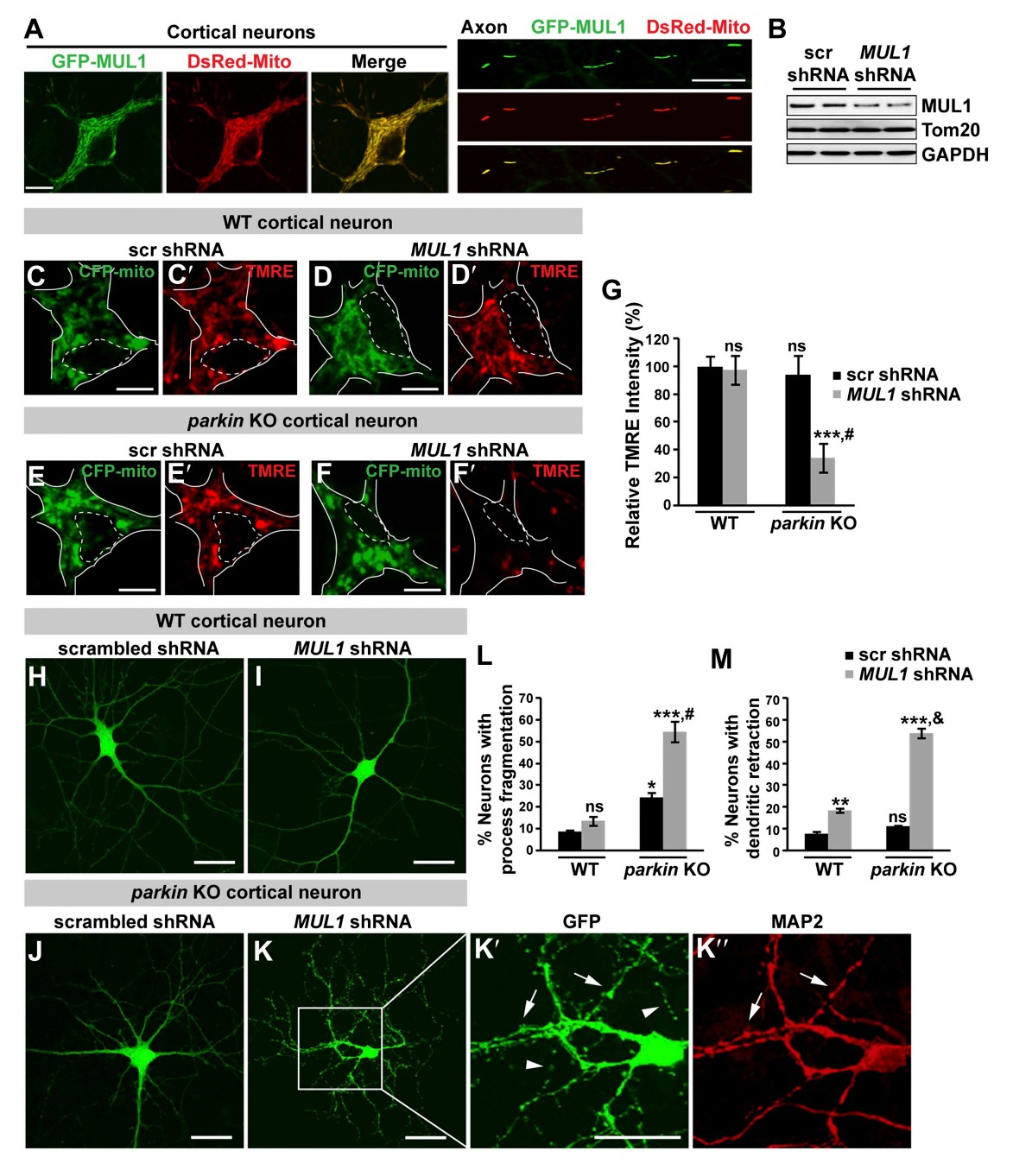

**Figure 8**. Loss of both *MUL1* and *parkin* aggravates mitochondrial damage and induces degeneration-like phenotypes in mouse cortical neurons. (**A**) MUL1 targets mitochondria in the cell bodies and axons of mouse primary cortical neurons. Neuronal mitochondria were labeled by DsRed-Mito or stained with an antibody against mitochondrial marker, TOM20 or Cytochrome C (*Figure 8—figure supplement 1*). (**B**) Levels of endogenous MUL1 in neurons transfected with scrambled or *MUL1* shRNA. Note that partial suppression of endogenous MUL1 may reflect relative low transfection rate (20%) in the neuronal culture. (**C**–**F'**) Mitochondria in live cortical neurons were co-labeled by expressing CFP-mito, which targets all mitochondria, and by loading fluorescent dye TMRE, which stains healthy mitochondria dependent upon membrane potential ($\Delta\psi_m$). Loading TMRE also labels mitochondria in glia in the culture. The edges of neuron cell bodies are marked with white solid lines, and the nuclei are outlined with white dashed lines. In contrast to other neurons, *parkin* knockout neurons with *MUL1* knockdown show reduced TMRE intensity (**F** and **F'**), indicating decreased $\Delta\psi_m$. Scale bars: 10 µm. (**G**) Quantification of relative TMRE intensity. TMRE intensity measured from each group of neurons was normalized to WT neurons transfected with scrambled shRNA. The data are shown as the means ± SEM from three experiments. (n ≥ 12 for each group). *** Significantly different from wild-type neurons transfected with scrambled shRNA, p<0.001. ns: not statistically significant (One-way ANOVA with Tukey's multiple

*Figure 8. Continued on next page*

*Figure 8. Continued*

comparisons test). # Significantly different from wild-type neurons transfected with *MUL1* shRNA and *parkin* KO neurons transfected with scrambled shRNA, p<0.001 and p<0.01, respectively (Two-way ANOVA with Tukey's multiple comparisons test). (**H–M**) *MUL1* knockdown in *parkin* KO neurons results in enhanced fragmentation of neurites. Representative wild-type (**H** and **I**) and *parkin* KO (**J–K″**) cortical neurons transfected with scrambled or *MUL1* shRNA and labeled with GFP (confirming transfection of shRNA and labeling axons and dendrites). (**K′–K″**) Higher magnification of a white box in **K** showing the soma and proximal dendrites labeled with an anti-MAP2 antibody (red). Arrows point to the GFP- and MAP2-labeled dendrites, and arrowheads indicate GFP-labeled but MAP2-negative fragmented axons. Scale bars: 20 µm. (**L** and **M**) Quantitative analysis showing enhanced process fragmentation (**L**) and dendritic retraction (**M**). The data are shown as the means ± SEM from three experiments (process fragmentation phenotype: n ≥ 115 for each genotype, dendritic retraction: n ≥ 127 for each phenotype). *, **, and *** Significantly different from wild-type neurons transfected with scrambled shRNA, p<0.05, p<0.01, and p<0.001, respectively. ns: not statistically significant (One-way ANOVA with Tukey's multiple comparisons test). # Significantly different from wild-type neurons transfected with *MUL1* shRNA and *parkin* KO neurons transfected with scrambled shRNA, both p<0.001. & Significantly different from wild-type neurons transfected with *MUL1* shRNA and *parkin* KO neurons transfected with scrambled shRNA, both p<0.001 (Two-way ANOVA with Tukey's multiple comparisons test).

The following figure supplements are available for figure 8:

**Figure supplement 1**. MUL1 localizes to mitochondria in mouse cortical neurons.

**Figure supplement 2**. *MUL1* knockdown increases Mfn2 levels in mouse cortical neurons.

Why do cells have multiple E3 ubiquitin ligases acting on a common target? Mfn is localized to the mitochondrial outer membrane (OM) (***Figure 9A***) and is a key molecule that regulates mitochondrial fusion in response to various cellular processes. Due to its importance, the level of Mfn is expected to be tightly regulated, and this may require several E3 ubiquitin ligases and deubiquitinases that respond to different stimuli (***Gegg et al., 2010***; ***Park et al., 2010***; ***Leboucher et al., 2012***; ***Lokireddy et al., 2012***; ***Anton et al., 2013***; ***Fu et al., 2013***). In the case of mitochondrial damage, Parkin translocates to depolarized mitochondria before it degrades Mfn (***Figure 9A***), thus preventing damaged mitochondria from fusing with healthy ones. As an E3 ligase anchored on the OM (***Li et al., 2008***; ***Figure 9A***), MUL1 is constantly present in the vicinity of Mfn, thus mediating Mfn clearance either constitutively or in a regulated manner in response to different stress signals. It is also possible that multiple E3 ligases work in a concerted way to ensure constant Mfn levels. In our study, CHX treatment leads to the stabilization of Mfn levels in HeLa cells lack of *MUL1*. However, steady-state Mfn levels in these cells are not strongly affected. This may result from the existence of other pathways for Mfn regulation, such as direct transcriptional feedback regulation on Mfn expression, activities of deubquitinases, and additional E3 ligases. Similar considerations may explain the viability and apparently mild phenotypes of *MUL1* mutant flies. More severe phenotypes may be uncovered in flies lacking *MUL1* in response to specific stresses that cannot be buffered by other components.

A recent study reports that *MUL1* promotes mitophagy, when muscle wasting is stimulated in mice (***Lokireddy et al., 2012***). To monitor mitophagy, this study measured mitochondrial DNA content and emission of a mitochondrial fluorescent protein that changes color in an acidic environment such as the lysosome (***Lokireddy et al., 2012***). However, since these methods do not directly visualize mitochondrial fate, it is possible that the observations may reflect early signs of mitochondrial dysfunction or turnover of the indicator protein, rather than clearance of the mitochondria. Also, it is unknown whether *MUL1* interacts with the *PINK1/parkin* pathway to regulate the mitochondrial clearance. Our results show that overexpression or lack of *MUL1* does not affect Parkin-mediated mitophagy induced by mitochondrial damage in HeLa cells. This further strengthens our hypothesis that *MUL1* acts in *PINK1/parkin*-independent pathway for regulating mitochondrial quality control.

Given our observations, it will be interesting to ask if human mutations in *mfn1 or mfn2* that decrease their abilities to be targeted for ubiquitin-dependent degradation, or mutations in *MUL1*, result in susceptibility for PD. It will also be interesting to see if polymorphisms in *MUL1* that affect *MUL1* expression levels or activity occur in PD patients. In this regard, it is worth noting that MUL1 forms a complex with VPS35 and VPS26 (***Braschi et al., 2010***). Since mutations in *VPS35* have been identified in multiple PD families (***Vilarino-Guell et al., 2011***; ***Zimprich et al., 2011***; ***Kumar et al., 2012***; ***Lesage et al., 2012***), it will be particularly interesting to determine if PD-associated mutations in *VPS35* have effects on *MUL1*-dependent degradation of Mfn. Finally, our observation that overexpression of *mfn* alone is sufficient to recapitulate key phenotypes associated with loss of *PINK1* or *parkin* suggests that inhibition of *mfn* may have important therapeutic potential for PD.

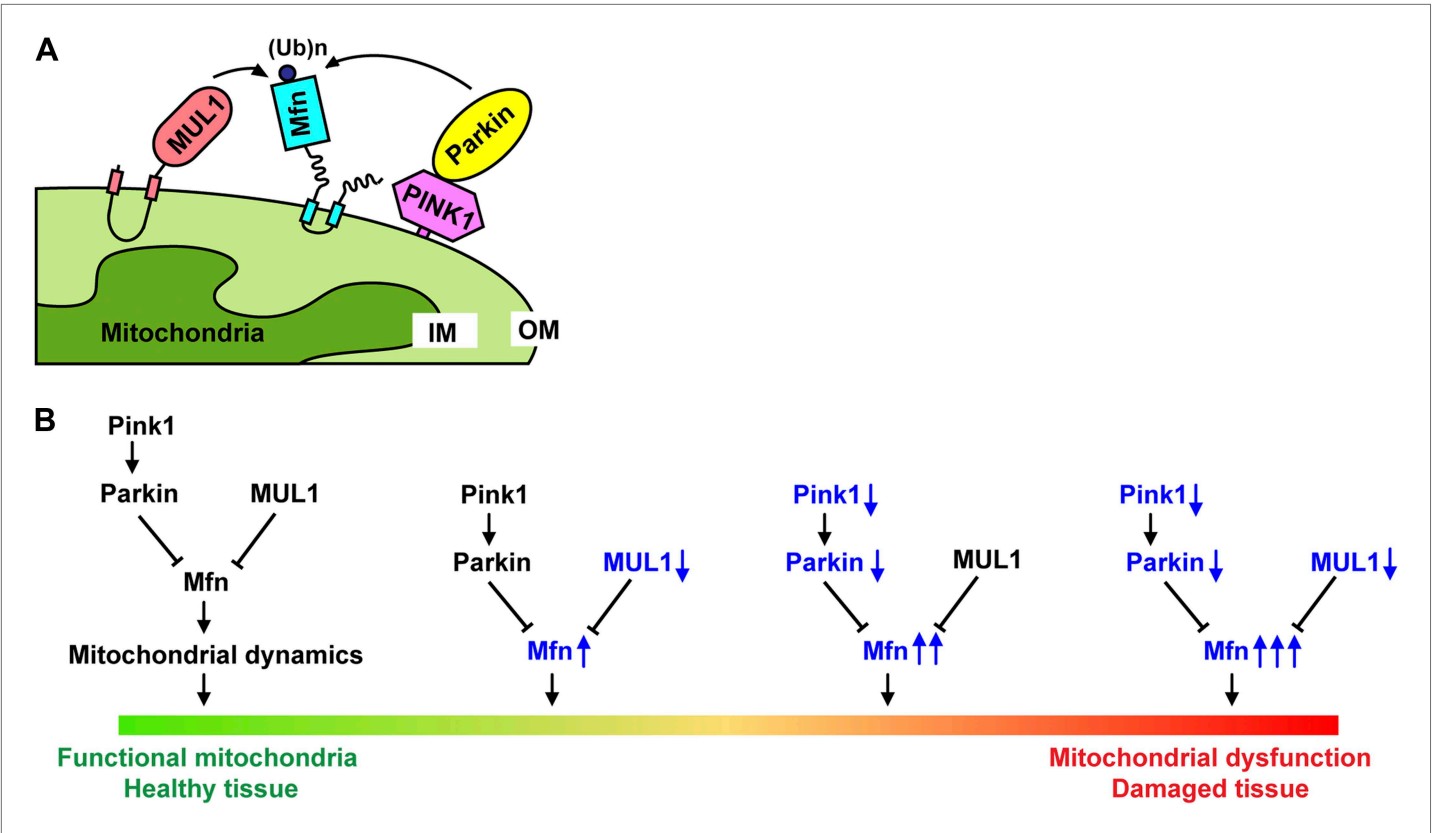

**Figure 9**. Models for how *MUL1* interacts with *PINK1/parkin*. (**A**) Schematic depictions of how MUL1, PINK1, Parkin, and Mfn interact in the mitochondria. In mammalian cells, upon mitochondrial damage (CCCP or antimycin A treatment), PINK1 is stabilized onto the mitochondrial OM of damaged mitochondria, with its kinase domain facing the cytosol (*Zhou et al., 2008*). PINK1 recruits Parkin onto the OM, either through direct phosphorylation or indirect interaction with other proteins (not depicted here) (*Jin and Youle, 2012*). Parkin then ubiquitinates multiple substrates on the OM, including Mfn. MUL1, a mitochondrial OM-anchored ligase with its RNF domain facing the cytosol, also mediates ubiquitination of Mitofusin. (**B**) The *PINK1/parkin* pathway and *MUL1* act in parallel to regulate *mfn*, and maintain mitochondrial function and tissue health. Reducing either *PINK1/parkin* or *MUL1* leads to increased levels of Mfn. Significant elevation of Mfn leads to mitochondrial dysfunction and tissue damage, similar to what is observed in *PINK1/parkin* mutants. Loss of both *PINK1/parkin* and *MUL1* leads to significantly higher Mfn levels, associated with severe mitochondrial dysfunction and tissue damage. OM: mitochondrial outer membrane; IM: mitochondrial inner membrane.

# Materials and methods

## Molecular biology and constructs

To generate UAS-*MUL1*, an EST clone from the *Drosophila* Genome Research Center (DGRC), AT15655, was subcloned into the UASt vector using EcoR1 and Xho1. The *Drosophila* MUL1 ligase-dead mutant (MUL1 LD) was generated by mutating H307 to A via site-specific mutagenesis (Stratagene QuikChange II XL Kit; Stratagene, La Jolla, CA). To generate UAS-*mfn*, the EST clone from DGRC, RE04414, was subcloned into the UASt vector. For UAS-*MUL1*-GFP, UAS-*mfn*-myc, and UAS-HA-*parkin*, each gene's coding region was fused to a different tag using the gateway cloning system (Invitrogen, Carlsbad, CA). To silence *MUL1* and *drp1*, the coding regions of *MUL1* and *drp1* transcripts were targeted using a synthetic microRNA-based technology (*Chen et al., 2006*; *Ganguly et al., 2008*). PCR products of these miRNA precursors were cloned into pUASt. To generate IFM-GAL4, the regulatory region of the *flightin* gene was used. All constructs made were confirmed by DNA sequencing. To map *MUL1* imprecise excision lines, breakpoints were determined by genomic PCR followed by DNA sequencing. pAC-*mfn*-Flag was a gift from Dr Alexander J Whitworth (*Ziviani et al., 2010*). Human MUL1 cDNA (BC010101) was purchased from ATCC and cloned into a pEGFP vector (Clontech, San Jose, CA) to generate GFP-*MUL1*. Flag-*MUL1* was generated by replacing the GFP tag with a Flag tag. Human MUL1 LD was generated by mutating H319 to A, which corresponds

to *Drosophila* MUL1 H307A. Human *MUL1* shRNA constructs were purchased from OriGene. The *MUL1* shRNA sequences are 5'-CTTCAAGTCCTGCGTCTTTCTGGAGTGTG-3' and 5'-GAAGGAGCT GTGCGGTCTGTTAAAG AAAC-3'.

## *Drosophila* genetics and strains

CaSpeR-HA-*drp1* flies were a gift from Dr Hugo J Bellen (*Verstreken et al., 2005*). *MUL1^EY12156^*, TRiP *parkin* RNAi, UAS-mitoGFP, Mef2-GAL4, OK6-GAL4 and TH-GAL4 flies were obtained from the Bloomington *Drosophila* Stock Center. *PINK1^5^*, *parkin^25^*, *dpk^21^*, UAS-*drp1* and UAS-*mfn* RNAi flies have been previously described (*Clark et al., 2006*; *Deng et al., 2008*). For experiments involving transgenic flies, constructs were injected into *w^1118^* and multiple independent fly lines were generated and analyzed (Rainbow Transgenic Flies, Inc.). The deletion mutant *MUL1^A6^* was generated by imprecise excision of *MUL1^EY12156^* using previously described methods (*Gross et al., 2013*). *Drosophila* strains were largely maintained in a 25°C humidified incubator.

## RNA isolation, cDNA synthesis, and quantitative PCR (qPCR)

RNA was isolated from whole flies using the Macherry-Nagel Nucleospin RNA II kit. cDNA synthesis was performed using the Clontech RNA to cDNA EcoDry Premix Kit, using a combination of Oligo-dT and random hexamer priming. Quantitative PCR was performed using the BioRadiTaq Fast Sybr Green enzyme mix, 10 µl reactions in triplicate, on a Roche Light Cycler 480. Standard curves were generated for *MUL1* and two control genes, *rpl32* and *eIF1α*. *Table 1*.

## Reverse transcription PCR (RT-PCR)

Total RNA was prepared as described above. RT-PCR was performed using Titanium One-Step RT-PCR Kit according to the manufacturer's instructions (Promega, Madison, WI). Primers used for RT-PCR are as follows in *Table 2*.

## Immunofluorescence and confocal microscopy

For analysis of muscle, thoraces of 1- to 2-day-old-adult flies were dissected and fixed in 4% paraformaldehyde in phosphate buffered saline (PBS). After thoraces were washed three times in PBS, muscle fibers were isolated and stained with rhodamine phalloidin (Invitrogen, 1:1000) in PBS+1% Triton X-100. For antibody staining, muscle fibers were permeabilized in PBS+0.1% Triton X-100, blocked in 5% normal goat serum in PBS, and incubated in primary and secondary antibodies diluted in 5% normal goat serum in PBS. For analysis of dopaminergic neurons, brains of 3-day-old male flies were dissected and fixed in 4% paraformaldehyde in PBS. Blocking, primary and secondary antibody staining were performed as described previously (*Yun et al., 2008*). To analyze mitochondria in salivary glands, salivary glands of third instar larvae were dissected, fixed in 4% paraformaldehyde in PBS, and stained with rhodamine phalloidin. The following primary antibodies were used: mouse anti-ATP Synthase (Mitosciences, Eugene, OR), chicken anti-HA (Millipore, Billerica, CA), mouse anti-Tyrosine Hydroxylase (Immunostar Hudson, WI). All images were taken on a Zeiss LSM5 confocal microscope.

## TUNEL assay

Adult male flies were aged for 5 days at 25°C. Thoraces of the flies were dissected and fixed in 4% paraformaldehyde in PBS. Muscle fibers were dissected and subsequently permeabilized and blocked in T-TBS-3% BSA (T-TBS: 0.1% Triton X-100, 50 mM Tris-Cl [pH 7.4], 188 mM NaCl). After blocking, TUNEL staining was carried out using an In Situ Cell Death Detection Kit according to the manufacturer's instructions (Roche, Switzerland).

## Embedding, sections, Toluidine blue staining, and transmission electron microscopy

Thoraces from 3-day-old male flies were dissected, fixed in paraformaldehyde/glutaraldehyde, post-fixed in osmium tetroxide, dehydrated in ethanol, and embedded in Epon. After polymerization of Epon, blocks were cut to generate 1.5-µm thick sections using a glass knife, or 80-nm thick sections

**Table 1.** Primer sequences for qPCR

| Primers | Sequence |
| --- | --- |
| MUL1-F | GCTATTGGTGAACTGGAGTTGGA |
| MUL1-R | AGCTTGAGTATCGTCGTTGTCTT |
| rpl32-F | TATGCTAAGCTGTCGCACAAATG |
| rpl32-R | GAACTTCTTGAATCCGGTGGGC |
| eIF1α-F | ACTTCGCAAGAAGGTGTGGATTA |
| eIF1α-R | GTACGTCTTCAGGTTCCTGGC |

**Table 2.** Primer sequences for RT-PCR

| Primers | Sequence |
| --- | --- |
| MUL1 RT-F | ACACGAATCCGT TGCACTG |
| MUL1 RT-R | GCTCGTAGTTGTCGTAGACC |

using a diamond knife on a microtome (Leica, Germany). Toluidine blue was used to stain 1.5-μm - thick tissue sections. Thin sections (80-nm thick) were stained with uranyl acetate and lead citrate, and examined using a JEOL 100C transmission electron microscope (UCLA Brain Research Institute Electron Microscopy Facility). At least six thoraces were examined in each sample.

## Quantification of mitochondrial number and size in salivary glands

Images were taken on a Zeiss LSM5 confocal microscope. Each cell in the image was outlined, and the outlined area was analyzed for mitochondrial number, average size and total area using the Analyze Particles function in ImageJ software (NIH). N = 8

## *Drosophila* lysate preparation and western blotting

Thoraces from adult flies or whole animals were homogenized in RIPA buffer containing protease inhibitors (Roche). Total protein concentration was measured using a Bradford assay kit (Bio-Rad, Hercules, CA), and the same amount of protein was loaded onto SDS-polyacryamide gels. The following primary antibodies were used for Western blots: mouse anti-myc (Millipore), mouse anti-HA (Millipore), mouse anti-Tubulin (Sigma, St. Louis, MO), rabbit anti-Actin (Sigma), mouse anti-Porin (mitosciences), and rabbit anti-Mfn (a generous gift from Dr Alexander J Whitworth).

## S2 cell culture, transfection, and RNAi treatment

S2 cells were cultured in Schneider's *Drosophila* Medium (Gibco, Grand Island, NY) with 10% fetal bovine serum (Invitrogen) and 1% penicillin/streptomycin (Invitrogen). Cells were seeded a day before transfection, and transfections were performed using the Effectene kit according to the manufacturer's recommendations (Qiagen, Valencia, CA). pAC-GAL4 was transfected along with UAS-Mfn-myc, UAS-HA-parkin, and UAS-MUL1-GFP for protein expression. UAS vector was used as empty vector. Cells were harvested 2 days after transfection. Double-stranded RNA (dsRNA) against coding regions of *GFP*, *PINK1*, *parkin*, *MUL1*, and *mfn* were generated using the T7 RiboMax express RNAi system (Promega). Primers that were used to generate dsRNAs are described below. S2 cells were seeded and treated with dsRNAs in serum-free medium for 40 min. After dsRNA treatment, complete medium was added to the culture, and the culture was incubated for 2–3 days. *Table 3*.

## Co-immunoprecipitation

S2 cells were lysed in RIPA buffer containing protease inhibitors (Roche), and Western blots were performed with 2% of lysates to check protein expression. Immunoprecipitations were performed with the rest of lysate using Dynabeads (Invitrogen) according to the manufacturer's instructions. Proteins bound to beads were eluted in SDS sample buffer, and Western blots were performed. Primary

**Table 3.** Primer sequences for the generation of dsRNA templates

| Primer | Sequence |
| --- | --- |
| GFP-F (control) | TAATACGACTCACTATAGGGTGAACCGCATCGAGCTGAA |
| GFP-R (control) | TAATACGACTCACTATAGGGACTTGTACAGCTCGTCCATG |
| PINK1-F | TAATACGACTCACTATAGGGAATGTGACTTCTCCAGCGA |
| PINK1-R | TAATACGACTCACTATAGGGTCGTAGCGTTTCATCAGCAG |
| parkin-F | TAATACGACTCACTATAGGGGTACGCAAAATGCTGGAGCT |
| parkin-R | TAATACGACTCACTATAGGGTAGAGGCTTGGAGGCTTCAT |
| MUL1 #1-F | TAATACGACTCACTATAGGGCCACCAAGTCCACGCTTATT |
| MUL1 #1-R | TAATACGACTCACTATAGGGTGATCCTGGGACAGAGTGTG |
| MUL1 #2-F | TAATACGACTCACTATAGGGGATTGTGAAGCTGCATGAGC |
| MUL1 #2-R | TAATACGACTCACTATAGGGAACACATGGTCGAAGAGGGA |

antibodies used include mouse anti-Myc (Millipore), rabbit anti-GFP (Invitrogen), rabbit anti-HA (Sigma), and rabbit anti-Actin (Sigma).

## In vivo ubiquitination assay in S2 cells

After treatment with dsRNA for 2 days, S2 cells were transfected with Mfn-Flag and incubated for 24 hr. Before harvest, cells were treated with the proteasome inhibitor MG132 (Millipore) for 4 hr. Cells were lysed and boiled in SDS lysis buffer (1% SDS, 150 mM NaCl, 10 mM Tris–HCl, pH 8.0) with protease inhibitors (Roche) for 10 min. Dilution buffer (10 mM Tris–HCl, pH 8.0, 150 mM NaCl, 2 mM EDTA, 1% Triton) was added, and immunoprecipitations were performed using mouse anti-Flag antibody (Sigma). After immunoprecipitations, Western blots were probed with mouse anti-ubiquitin (Covance). Mouse anti-FK1 (Enzo Life Sciences, Farmingdale, NY) and anti-FLAG (Sigma) antibodies were used.

## Protein purification and in vitro ubiquitination assay

For in vitro ubiquitination assay, the glutathione S-transferase (GST)-tagged expression vectors pGex-MUL1 and pGEX-MUL1 LD were generated. GST fusion proteins (GST-MUL1 and GST-MUL1 LD) were expressed in *E. coli* and purified from inclusion body. The in vitro ubiquitination assay was performed using the following buffer: 25 mM Tris (pH 7.5), 5 mM MgCl2, 100 mM NaCl, 1 mM DTT, 0.05 mM MG132, 2 mM ATP, and 0.125 µg/µl Ubiquitin, with E1 (Rabbit, 0.5 µg/ml), E2 (extract from *E. coli* expressing UbcH5C), and presence or absence of GST-MUL1, or GST-MUL1 LD (as indicated). Reaction mixtures were incubated at 30°C for 2 hr, and reactions were terminated by boiling in SDS loading buffer.

## Mammalian cell culture, transfection, and western blotting

HeLa cells that did or did not overexpress *parkin* were generous gifts from Dr David C Chan (*Chan et al., 2011*). Cells were cultured in Dulbeco's modified Eagle's medium (DMEM, Gibco) containing 10% fetal bovine serum (Invitrogen) and 1% penicillin/streptomycin (Invitrogen). Cells were plated a day before transfections, and transfections were performed using the Effectene kit (Qiagen) or X-tremeGENE 9 DNA Transfection Reagent (Roche) according to the manufacturer's recommendations. After transfections, Z-VAD-FMK (Santa Cruz Biotechnology, Santa Cruz, CA) was added to cultures every 24 hr to inhibit apoptosis. Cells were harvested 48 hr later and lysed in RIPA buffer containing protease inhibitors (Roche). Western blots were performed with the following primary antibodies: rabbit anti-human MUL1 (Sigma), mouse anti-Mfn1 (Abcam), mouse anti-Mfn2 (Abcam), rabbit anti-Actin (Sigma), and mouse anti-Porin (Mitosciences).

## Parkin-mediated mitophagy assays

HeLa cells that did or did not stably express *MUL1* shRNA were seeded in chamber slides, respectively, and transfected with YFP-Parkin one day later. 24 hrs after transfection, cells were treated with DMSO or 40 µg/ml Antimycin A (Sigma) for 1.5, 3, 24, or 48 hrs as indicated to dissipate mitochondrial membrane potential. For *MUL1* overexpression, HeLa cells stably expressing YFP-Parkin and mitoRFP (a kind gift from Dr. Mark R Cookson) were seeded and transfected with Myc-MUL1 1 day later. Cells were treated with DMSO or 80 µg/ml Antimycin A for 1.5 or 3 hrs. After treatment of Antimycin A, cells were fixed in 10% Formalin solution (Sigma), permeabilized with 0.1% Triton X-100 in PBS, and blocked in PBS containing 5% fetal bovine serum. Primary and secondary antibody staining were performed in 5% fetal bovine serum + PBS. The following primary antibodies were used: mouse anti-Tom20 (BD), mouse anti-Flag (Sigma), rabbit anti-GFP (Invitrogen), and rabbit anti-Parkin (Abcam, Cambridge, MA). More than 100 cells for each experiment were counted for quantification, and the experiments were repeated twice. PINK1 knockout cells were a generous gift from Dr. Richard Youle. Western blot analysis confirmed that there is no PINK1 expression in *PINK1* knockout cells (*Narendra et al., 2013*; personal communication).

## Generation of *MUL1* knockout (*MUL1−/−*) HeLa cells using CRISPR/Cas 9 system

*MUL1* knockout HeLa cells were generated using the CRISPR/Cas system as previously described (*Cong et al., 2013*). Briefly, *MUL1* targeting sequence 5'-GCCGCCGTCA TGGAGAGCGG-3' was inserted into pX330-U6-Chimeric_BB-CBh-hSpCas9 (Addgene). HeLa cells were seeded a day before transfection. Cells were transfected with the construct using X-tremeGENE 9 DNA transfection

reagent (Roche) following manufacturer's instructions. 2 days after the transfection, cells were diluted and split into 48 well plates. Each colony was screened for deletions in *MUL1* by PCR and sequencing using a set of primers 5'-CGCCTCGAACCTGACACATAATAGG-3' and 5'-GTCTGTAAAGCAAGGAGTG GAGTGG-3'. Two *MUL1* knockout cells were isolated. Both *MUL1* knockout cells have deletions including the start codon of *MUL1*, one with 228 base pair deletion and another with 8 base pair deletion. Both deletions result in frame shift and early termination of protein translation. Further western blot analysis using two different anti-MUL1 antibodies (Sigma) confirmed that there is no MUL1 expression in *MUL1*−/− cells.

## Protein turnover

HeLa cells that express scrambled shRNA or *MUL1* shRNA were treated with cycloheximide (Sigma) for 0, 2, 4, 6 hrs. After cycloheximide treatment, cells were harvested and lysed. Protein concentration of each lysate was determined by Bradford assay (Bio-Rad), and an equal amount of total protein was subjected to Western blot. Blots were probed with anti-Mfn1 (Abcam), anti-Mfn2 (Abcam) and Actin (Sigma) antibodies. Levels of Mfn2 and Actin were quantified using ImageJ.

## Mouse cortical neuronal culturing, transfection, and immunocytochemistry

Animal care and use were carried out in accordance with NIH guidelines, NIH Manual 3040-2, Guide for the Care and Use of Laboratory Animals (National Research Council), Institutional Animal Care and Use Committee Guidebook (ARENA and OLAW) and approved by the NIH, NINDS/NIDCD Animal Care and Use Committee on 3/5/2012 (ASP# 1303-9).

The work and submission for publication was approved by the Intramural Program of NINDS, NIH. Dissection of embryonic mouse brains and isolation of cortical neurons and plating were designed to be very quick with minimal enzymatic, mechanical, chemical, and oxidative damage, as described by *Cai et al. (2012)*. Cortices were dissected from E18-19 mouse embryos. Cortical neurons were dissociated by papain (Worthington) and plated on glial beds at a density of 50,000 cells per cm$^2$ on polyornithine (Sigma) and Matrigel (BD Biosciences)-coated coverslips. Neurons were grown overnight in plating medium (5% FBS, insulin, glutamate, G5, 1 x B27 and beta-mercaptoethanol) supplemented with 100 x L-glutamine in Neurobasal (Invitrogen). Starting at DIV2, cultures were maintained in conditioned medium with half-feed changes of neuronal feed (1 × B27, 100 × L-glutamine and beta-mercaptoethanol in Neurobasal) every 3 days. Neurons were transfected with various constructs at DIV7-8 using calcium phosphate and processed for immunocytochemistry 72 hr (DIV10-11) post transfection.

For immunostaining, cultured cells were fixed with 4% formaldehyde (Electron Microscopy Sciences) and 4% sucrose (Sigma) in 1X phosphate-buffered saline (PBS) at 4°C for 30 min, washed three times with PBS for 5 min each, and then incubated in 0.2% saponin, 5% normal goat serum (NGS), and 2% bovine serum albumin (BSA) in PBS for 1 hr. Fixed cultures were incubated with primary antibodies in PBS with 1% BSA and 0.05% saponin at 4°C overnight. Cells were washed four times with PBS at RT for 5 min each, incubated with secondary fluorescent antibodies at 1:400 dilution in PBS with 1% BSA and 0.05% saponin for 60 min, re-washed with PBS, and then mounted with Fluoro-Gel anti-fade mounting medium (EMS) for imaging. Sources of antibodies are as follow: polyclonal antibodies against TOM20 (Santa Cruz), MUL1 (Sigma), Mfn2 (Cell Signaling, Danvers, MA); monoclonal antibodies against MAP2 (Millipore), GAPDH (Research Diagnostic, Hackensack, NJ), CytC (BD Biosciences, San Jose, CA); Alexafluor 546 and 633-conjugated secondary antibodies (Invitrogen). Confocal images were obtained using an Olympus Fluoview FV1000 microscope, oil immersion 63X objective (NA-1.45) with sequential-acquisition settings. Images were acquired using the same settings below saturation at a resolution of 1024X1024 pixels (12 bit). Z stacks were acquired using a step size of 0.37 μm from top to bottom, and brightest point projections were made. For quantification, data were obtained from at least three independent experiments and the number of cells used for quantification is indicated in the figures. All statistical analyses were performed using One-way or Two-way ANOVA with Tukey's multiple comparison test and are presented as mean ± SEM.

## TMRE (tetramethyl rhodamine ethyl ester) staining

To access mitochondrial potential on a single cell basis, mature cortical neurons DIV (10–11), both from wild-type (WT) and *parkin* knockout (KO), were incubated with the cationic lipophilic compound TMRE (50 nM) for 20 min in a 37°C CO$_2$ incubator. Post treatment, cells were washed three times with

imaging media and mounted for imaging. Confocal images were obtained using an Olympus confocal oil immersion 63x objective with the sequential-acquisition setting. The images were acquired within 30 min. For fluorescent quantification, image acquisition settings were below saturation at a resolution of 1024 × 1024 pixels (12 bit). Five to six sections were taken from the top-to-bottom of the specimen and brightest point projections were made. Morphometric measurements were performed using NIH ImageJ. Measured data were imported into Excel software for analysis. The thresholds in all images were set to similar levels. Fluorescence intensity of TMRE was expressed in corrected total cell fluorescence (CTCF) values. The mean intensity of TMRE in the soma of each group was normalized as a percentile ratio relative to that in WT cells expressing scrambled shRNA. Data were obtained from at least three independent experiments and the number of cells used for quantification is indicated in the figures. All statistical analyses were performed using one-way or two-way ANOVA with Tukey's multiple comparisons test and are presented as mean ± SEM.

## Acknowledgements

We thank H Bellen, D Chan, M Cookson, A Whitworth, and R Youle for reagents, H Mcbride for communicating unpublished results, H Huang and BA Hay for generating IFM-Gal4 lines, H Deng, H Huang, Y Sun, and B Al-Anzi for technical assistance, BA Hay, L Leung, P Patel, and CY Lee for comments on the manuscript, and F Laski, L Dreier and N Freimer for use of their equipment. This work was supported by a UCLA Dissertation Fellowship to JY, the NIH-DBT Khorana Nirenberg Scholarship to RP, the Chinese Scholarship Council Fellowship to HY, the intramural research program of NIH/NINDS to Z-HS., and NIH (R01, K02, P01), the McKnight Neuroscience Foundation, the American Parkinson's Disease Association, the Klingenstein Fellowship Award in the Neurosciences, the Kenneth Glenn Family Foundation, and the Ellison Medical Foundation Senior Scholar Award to MG.

## Additional information

### Funding

| Funder | Grant reference number | Author |
|---|---|---|
| McKnight Endowment Fund for Neuroscience | | Ming Guo |
| National Institutes of Health | R01, K02, P01 | Ming Guo |
| Ellison Medical Foundation | Senior Scholar Award | Ming Guo |
| Esther A. and Joseph Klingenstein Fund | Klingenstein-Simons Fellowship Award in Neurosciences | Ming Guo |
| The American Parkinson Disease Association | | Ming Guo |
| Glenn Family Foundation | | Ming Guo |
| National Institute of Neurological Disorders and Stroke | Intramural research program | Zu-Hang Sheng |
| The Chinese Scholarship Council Fellowship | | Huan Yang |
| NIH-DBT Khorana Nirenberg Scholarship | | Rajat Puri |
| UCLA Dissertation Fellowship | | Jina Yun |

The funders had no role in study design, data collection and interpretation, or the decision to submit the work for publication.

### Author contributions

JY, Conception and design, Acquisition of data, Analysis and interpretation of data, Drafting or revising the article; RP, HY, Conception and design, Acquisition of data, Analysis and interpretation of data, Drafting or revising the article; MAL, CW, Acquisition of data, Analysis and interpretation of data; Z-HS, MG, Conception and design, Analysis and interpretation of data, Drafting or revising the article.

## Ethics

Animal experimentation: Animal care and use were carried out in accordance with NIH guidelines, NIH Manual 3040-2, Guide for the Care and Use of Laboratory Animals (National Research Council), Institutional Animal Care and Use Committee Guidebook (ARENA and OLAW) and approved by the NIH, NINDS/NIDCD Animal Care and Use Committee on 3/5/2012 (ASP# 1303-9).The work and submission for publication was approved by the Intramural Program of NINDS, NIH.

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
