## [Decision Letter]

Thank you for sending your work entitled “*MUL1* compensates for loss of *PINK1/parkin* in maintaining mitochondrial integrity” for consideration at *eLife*. Your article has been favorably evaluated by a Senior editor and 3 reviewers, including member of our Board of Reviewing Editors Ray Deshaies and reviewer Hugo Bellen, who agreed to reveal their identities.

The Reviewing editor and the other reviewers discussed their comments before we reached this decision, and the Reviewing editor has assembled the following comments to help you prepare a revised submission.

The manuscript from the Guo lab provides evidence that in *Drosophila*, the ubiquitin ligase *MUL1* acts in a pathway parallel to *PINK* and *parkin* to maintain low levels of mitofusin protein. These findings are of considerable interest to the biological function of parkin and the consequences of its mutation in neurodegenerative disease, and are likely to be of interest to a general audience. In evaluating the claims made by the authors, the reviewers were of the consensus opinion that several modifications to the experimental data are needed to bring the claims in-line with the evidence provided to support them, as noted below:

1) Figure 3 appears to be the exact same blot as the first four lanes of Figure 5R. This is not acceptable. Since there is a quantification of the gel in Figure 3 beneath the gel with error bars, it should be possible to change the western blot in Figure 3 to one of the three repeat gels so it is not exactly the same gel shown as Figure 5R. Alternatively, the full figure could be shown in Figure 3 and the authors can refer back to the last 2 lanes when describing this result in the context of the rest of Figure 5.

2) Test whether endogenous Mfn co-IPs with endogenous *MUL1*. This would strengthen the fairly weak conclusion of a “direct” activity of *MUL1* on Mfn.

3) The lack of mitochondrial fusion in their drp1 RNAi in Figure 4P needs to be omitted or addressed in much more detail. If the authors with to sustain their claims regarding Drp1, they need to provide biochemical data to document that Drp1 is strongly silenced, and if it is not they would need to use a strong mutant or more potent RNAi to achieve a compelling loss-of-function.

4) It is surprising that both *Parkin/Pink* and *Mul1* pathways ubiquitinate mitofusin but Mul1 is not required for mitophagy in their model. They should test mitophagy at intermediate points before all the mitochondria disappear (e.g., 6,12, and 18 hr). The present data do not exclude the possibility that mitophagy is delayed in *MUL1* mutant cells. The authors should also test whether antimycin A can trigger mitophagy in cells that overexpress *MUL1*, in the presence and absence of *Parkin* or *PINK*. For both of the experiments described above, it would be desirable to have a more quantitative read-out, such as immunoblots of mitochondrial matrix proteins.

5) No data are presented to confirm that SUMO RNAi was effective in *PINK1* mutants suppressed by *MUL1* OE. Moreover, the data presented for wild type flies treated with SUMO RNAi is not quantitative. Given that this observation brings into question a published finding from the McBride lab, either the evidence for SUMO loss-of-function should be strengthened (e.g., by including blotting data) or the data should be removed.

In addition to these modifications to the data, there are a number of issues noted below that need to be addressed, but this can be done adequately by modifications to the text.

6) There is some concern among the reviewers concerning how the findings are presented. For example, the authors say that *MUL1* compensates for loss of *Parkin* and that *MUL1* ensures mitochondrial integrity and function. This seems to be an excessively 'parkin-ocentric' viewpoint, since it seems unlikely that MUL1 exists solely to compensate for the potential loss of *PINK* or *parkin*. The title and abstract should be modified to emphasize the parallel nature of the *PINK/parkin* and *MUL1* pathways.

7) In the Results section the authors refer to a “ligase-dead” version of MUL1, but there is no citation or data to establish why this mutant should be ligase-dead, or that it is in fact ligase-dead.

8) Provide more support for the claim in the Results section that visualization of mitochondrial ATPase is independent of mitochondrial import? Does assembly of ATPase holoenzyme rely on functional import pathway?

9) Report the Y-axis on the graph in Figure 5 as Mfn:actin ratio. Essentially all of the “increase” in relative Mfn in the double mutant is not due to an increase in Mfn at all, but rather a decrease in actin. Is it possible to normalize the samples in some other way (total protein or some other housekeeping protein besides actin)?

10) There was some divergence of opinion among the reviewers regarding the lack of a clear phenotype for mutant flies lacking *MUL1* (see comment #1 of reviewer 3). The authors should at least comment on whether knockdown/knockout of MUL1 has any noticeable effect on mitochondrial morphology. The authors should also discuss possible physiological roles of *MUL1* that could be further tested. Related to this, the authors should expand on their model in the discussion to explain that Mitofusin degradation presumably needs to reach a threshold to observe a phenotype like that seen for *Pink1/Parkin* mutants.

11) Figure 6: despite the fact that both Mfn1 and Mfn2 are stabilized upon depletion or deletion of MUL1, the steady-state levels of these proteins are either not affected or barely affected. The authors should comment on this.

In addition to these points, each reviewer raised additional points for which a consensus opinion did not emerge regarding how they should be dealt with. In these cases it is left to the discretion of the authors as to how to respond. These are listed below:

*Reviewer 1*:

1) The authors state, “...supporting the idea that *MUL1* suppresses *PINK1* mutant phenotypes through reduction of Mfn levels”. Although this is shown in wild type animals (Figure 3), it was not shown in *PINK1* mutant. While this may seem like a detail one could argue it is a relatively important omission since it is at the core of their interpretation of how *MUL1* OE is exerting its effect.

2) Figure 4M, O, S, U: does overexpression of parkin or MUL1 proteins reduce the level of overexpression of mitofusin protein?

3) Figure 5: why don't the *PINK1* and *parkin* mutants shown here look much better (i.e., much less defective) than those shown in multiple panels of Figures 1, 2 and 4?

4) Figure 8: what effect do these various manipulations have on mitofusin levels in cortical neurons?

*Reviewer 2*:

This manuscript concludes that Mul1, a mitochondrial E3 ligase, ubiquitinates and causes proteosomal degradation of mitofusin in Drosophila and mammalian cells. Furthermore, *Mul1* loss exacerbates the phenotype of Parkin loss. The loss of *Mul1* is clearly shown in Figure 5 to make the loss of *Parkin/Pink1* mutant flies worse, which indicates they work by different mechanisms. However, this is not surprising – many mutations in flies will cause stress by different paths and exacerbate one another's phenotype. Therefore, the key issue is whether or not both *Mul1* and *Parkin* function by inducing the ubiquitination and elimination of mitofusin. *Mul1* decreases the expression level of Mfn1 but this may be indirect and cell free experiments could conclusively show *Mul1* ubiquitinates Mfn. However, cell free experiments seems to be beyond the scope of this manuscript. Why would increased Mfn level lead to the mitochondrial and cellular defects? Is this through decreased mitophagy? If not mitophagy what? MUL1 is not likely expressed simply to statically reduce the steady state level of Mfn. How MUL1 is regulated and why and when it ubiquitinates MFn (if it is direct) remains unexplored. Although there are clearly mysteries here, this is a very thorough genetic study with intriguing conclusions in need of only minor corrections.

*Reviewer 3*:

MUL1 compensates for loss of *PINK1/parkin* in maintaining mitochondrial integrity. In the present manuscript, Yun et al. uncover a parallel pathway to PINK1/parkin acting through MUL1. Both pathways converge on a shared target mitofusin to maintain mitochondrial integrity in both *Drosophila* and mammals. Recently, MUL1 was identified as a novel regulator of mitochondrial fission by activation of Drp1 through sumolyation and degradation of MFN1 and MFN2 through ubiquitination. In this manuscript the authors provide genetic evidence that MUL1 is required for mitofusin turnover in *Drosophila* and human cell lines. Moreover, MUL1 overexpression can suppress phenotypes associated with pink and parkin mutants through mitofusin degradation, suggesting a compensatory role for MUL1. As *Pink1/Parkin* also ubiquitinate mitofusin, they decided to test if MUL1 and *Pink1/Parkin* act in the same pathway. Double mutant of MUL1 with either *PINK1* or *parkin* have a more severe phenotype than the single mutants, which suggests that MUL1 and *Pink1/Parkin* act in parallel pathways. I believe the manuscript suitable for *eLife* if the authors address these concerns:

1) The subtle phenotypes associated with *MUL1* mutants suggest a compensatory role of this pathway. The authors should highlight the importance of *MUL1* in physiological conditions. Can the authors document, either by loss of one copy of *Pink1* or *Parkin* or by stressing mitochondria, a stronger phenotype in *MUL1* mutants. They should test life span or fertility in *MUL1* mutants under normal and stressed conditions to provide a comparison with *pink/parkin* mutant phenotypes.

[Editors' note: further clarifications were requested prior to acceptance, as described below.]

Thank you for resubmitting your work entitled “*MUL1* acts in parallel to the *PINK1/parkin* pathway in regulating mitofusin and compensating for loss of *PINK1/parkin*” for further consideration at eLife. Your revised article has been favorably evaluated by a Senior editor, a member of the Board of Reviewing Editors, and the original reviewers. The manuscript has been improved but there are some remaining issues that need to be addressed. The authors have addressed most of the comments that were raised in the original review, with one notable exception. The authors have not shown that there is a direct physical interaction between the endogenous Mul1 and Mfn proteins. The authors state that this is not possible, because there is no antibody against *Drosophila* Mul1 and the antibody against mammalian Mul1 does not work for IP. However, there are antibodies available against mammalian Mfn that have been reported to work for immunoprecipitation. This experiment should be done, as originally requested, or the authors should present a strong argument for why it cannot be done.

---

## [Author Response]

*1)*
Figure 3
*appears to be the exact same blot as the first four lanes of Figure 5R. This is not acceptable. Since there is a quantification of the gel in*
Figure 3
*beneath the gel with error bars, it should be possible to change the western blot in*
Figure 3
*to one of the three repeat gels so it is not exactly the same gel shown as Figure 5R. Alternatively, the full figure could be shown in*
Figure 3
*and the authors can refer back to the last 2 lanes when describing this result in the context of the rest of*
Figure 5.

We agree with this point and thank the reviewer for pointing this out. We now added Figure 3 with one of the repeat gels.

*2) Test whether endogenous Mfn co-IPs with endogenous* MUL1*. This would strengthen the fairly weak conclusion of a “direct” activity of* MUL1 *on Mfn*.

This is a very reasonable experiment and we would like to be able to do this as well. However, there is no anti-*Drosophila MUL1* antibody available, which prevents us from carrying out this experiment in flies. We also tried the experiments using mammalian anti-*MUL1* antibodies in mammalian cells, but found the antibody not suitable for immunoprecipitation of endogenous proteins.

It is important to note that we have not made any strong argument that the interaction of MUL1 and Mfn is “direct”. In the Discussion, we state “These observations suggest that MUL1 may directly ubiquitinate Mfn, leading to its degradation. Alternatively, *MUL1* may act via an intermediary that promotes Mfn ubiquitination and degradation”.*3) The lack of mitochondrial fusion in their drp1 RNAi in Figure 4P needs to be omitted or addressed in much more detail. If the authors with to sustain their claims regarding Drp1, they need to provide biochemical data to document that Drp1 is strongly silenced, and if it is not they would need to use a strong mutant or more potent RNAi to achieve a compelling loss-of-function.* This is a great point.

In this revision, we have used flies with two different *drp1* null alleles (*drp1*^*1*^and *drp1*^*2*^, as previously reported by Vestreken et al.) instead of flies with *drp1* RNAi. Consistent with the RNAi results, flies with a complete loss-of-function of *drp1* do not show any TUNEL-positive cell death, which is distinct from lack of *PINK1* or *parkin*, or *mfn* overexpression. These panels are included as Figure 4V and Y.

*4) It is surprising that both* Parkin/Pink *and Mul1 pathways ubiquitinate mitofusin but Mul1 is not required for mitophagy in their model. They should test mitophagy at intermediate points before all the mitochondria disappear (e.g., 6,12, and 18 hr). The present data do not exclude the possibility that mitophagy is delayed in* MUL1 *mutant cells. The authors should also test whether antimycin A can trigger mitophagy in cells that overexpress* MUL1*, in the presence and absence of Parkin or PINK. For both of the experiments described above, it would be desirable to have a more quantitative read-out, such as immunoblots of mitochondrial matrix proteins*.

Following the reviewers’ suggestion, we have now tested 4 additional time points (6, 12, 16 and 20 hours) in addition to 24 hours for mitophagy. We were unable to see any differences between *MUL1* null cells and wildtype cells. We have modified the text accordingly.

Regarding testing whether antimycin A can trigger mitophagy in cells that overexpress *MUL1*, in the presence and absence of *Parkin or PINK1*, we think that it is an important question. However, we believe that it is beyond the scope of this work. The focus of the mammalian work in this manuscript is to address the requirement and physiological roles of MUL1 in mitophagy, when expressed at physiological levels. We will address whether MUL1 is sufficient to induce mitophagy in future work. However, if Reviewers feel strongly that these experiments are absolutely required for publication, we will of course carry out the experiments and report back to you.

*5) No data are presented to confirm that SUMO RNAi was effective in* PINK1 *mutants suppressed by* MUL1 *OE. Moreover, the data presented for wild type flies treated with SUMO RNAi is not quantitative. Given that this observation brings into question a published finding from the McBride lab, either the evidence for SUMO loss-of-function should be strengthened (e.g., by including blotting data) or the data should be removed*.

This is an excellent point. After consideration, we decided to remove this part.

*6) There is some concern among the reviewers concerning how the findings are presented. For example, the authors say that* MUL1 *compensates for loss of* Parkin *and that* MUL1 *ensures mitochondrial integrity and function. This seems to be an excessively 'parkin-ocentric' viewpoint, since it seems unlikely that* MUL1 *exists solely to compensate for the potential loss of* PINK *or parkin. The title and abstract should be modified to emphasize the parallel nature of the* PINK/parkin *and* MUL1 *pathways*.

We see the reviewers’ point and have modified the title and text. The revised title is “*MUL1* acts in parallel to the *PINK1/parkin* pathway in regulating *mitofusin* and compensating for loss of *PINK1/parkin*”. The revised Discussion also contains new paragraphs that discuss the role of *MUL1* in the context of its phenotypes.

*7) In the Results section the authors refer to a “ligase-dead” version of* MUL1*, but there is no citation or data to establish why this mutant should be ligase-dead, or that it is in fact ligase-dead*.

We thank reviewers for pointing this out. We have added the citation accordingly. In addition, we have added data from in vitro ubiquitination assays suggesting that *MUL1*, but not ligase-dead version, can self-ubiquitinate as Figure 1—figure supplement 1.

*8) Provide more support for the claim in the Results section that visualization of mitochondrial ATPase is independent of mitochondrial import? Does assembly of ATPase holoenzyme rely on functional import pathway*?

We see reviewers’ point and deleted the sentence in the text. We have clarified it by adding the following sentence to the figure legend: “Instead of using mitoGFP, we utilized anti-ATPase antibody, which allows better visualization of the enhancement phenotypes seen with double mutants.”

*9) Report the Y-axis on the graph in*
Figure 5
*as Mfn:actin ratio. Essentially all of the “increase” in relative Mfn in the double mutant is not due to an increase in Mfn at all, but rather a decrease in actin. Is it possible to normalize the samples in some other way (total protein or some other housekeeping protein besides actin)*?

This is a great point raised by reviewers. We repeated the experiment and measured protein concentration. We also used Tubulin in addition to Actin to normalize the samples. The updated figure is shown as Figure 5.

*10) There was some divergence of opinion among the reviewers regarding the lack of a clear phenotype for mutant flies lacking* MUL1 *(see comment #1 of reviewer 3). The authors should at least comment on whether knockdown/knockout of* MUL1 *has any noticeable effect on mitochondrial morphology. The authors should also discuss possible physiological roles of* MUL1 *that could be further tested. Related to this, the authors should expand on their model in the discussion to explain that Mitofusin degradation presumably needs to reach a threshold to observe a phenotype like that seen for* Pink1/Parkin *mutants*.

The effects of *MUL1*on mitochondrial morphology were in fact shown in Figure 2. MUL1 overexpression results in small and fragmented mitochondria. Loss of MUL1 is viable, albeit with mild mitochondrial elongation. It is possible that Mfn levels are kept constant by several different pathways including *MUL1* and the *PINK1/parkin* pathway. Due to the redundancy of regulation, loss of *MUL1* only causes mild phenotypes. It is also possible that certain stresses can lead to more severe phenotypes in *MUL1* null flies. We have included this part in our Discussion.

This may result from the existence of other pathways for Mfn regulation, such as direct transcriptional feedback regulation on Mfn expression, activities of deubquitinases, and additional E3 ligases). Similar considerations may explain the viability and apparent mild phenotypes of *MUL1* mutant flies. More severe phenotypes may be uncovered in flies lacking *MUL1* in response to specific stresses that cannot be buffered by other components.

It is an excellent idea to expand on our model to discuss the phenotypic threshold and mfn levels, and we appreciate the Reviewers for this suggestion. We have included the following sentences in our Discussion.

We hypothesize that the increase in the Mfn level needs to reach a threshold, such as that observed in the *PINK1/parkin* mutant backgrounds, but not in the *MUL1* null background, in order for overt muscle cell degeneration to occur (Figure 9).

*11)*
Figure 6*: despite the fact that both Mfn1 and Mfn2 are stabilized upon depletion or deletion of* MUL1*, the steady-state levels of these proteins are either not affected or barely affected. The authors should comment on this*.

This is a great suggestion and we added a paragraph in Discussion to address this.

Why do cells have multiple E3 ubiquitin ligases acting on a common target? Mfn is localized to the mitochondrial outer membrane (OM) (Figure 9 a key molecule that regulates mitochondrial fusion in response to various cellular processes. Due to its importance, the level of Mfn is expected to be tightly regulated, and this may require several E3 ubiquitin ligases and deubiquitinases which respond to different stimuli {[50] #5648;[35] #5649;[19] #5650;[22] #1310;[39] #1337;[2] #5651}. In the case of mitochondrial damage, *Parkin* translocates to depolarized mitochondria before it degrades Mfn (Figure 9), thus preventing damaged mitochondria from fusing with healthy ones. As an E3 ligase anchored on the OM (38) (Figure 9), *MUL1* is constantly present in the vicinity of Mfn, thus mediating Mfn clearance either constitutively or in a regulated manner in response to different stress signals. It is also possible that multiple E3 ligases work in a concerted way to ensure constant Mfn levels.

In our study, CHX treatment leads to the stabilization of Mfn levels in HeLa cells lack of *MUL1*. However, steady-state Mfn levels in these cells are not strongly affected. This may result from the existence of other pathways for Mfn regulation, such as direct transcriptional feedback regulation on Mfn expression, activities of deubquitinases, and additional E3 ligases. Similar considerations may explain the viability and apparent mild phenotypes of *MUL1* mutant flies. More severe phenotypes may be uncovered in flies lacking *MUL1* in response to specific stresses that cannot be buffered by other components.

*In addition to these points, each reviewer raised additional points for which a consensus opinion did not emerge regarding how they should be dealt with. In these cases it is left to the discretion of the authors as to how to respond*.

Reviewer 1:

*1) The authors state, “...supporting the idea that* MUL1 *suppresses* PINK1 *mutant phenotypes through reduction of Mfn levels”. Although this is shown in wild type animals (*Figure 3*), it was not shown in* PINK1 *mutant. While this may seem like a detail one could argue it is a relatively important omission since it is at the core of their interpretation of how* MUL1 *OE is exerting its effect*.

We fully agree with the point raised by Reviewer 1. In the revision, we added Figure 3 to show that the increased Mfn levels in PINK1 mutants are reduced when MUL1 is overexpressed. This further strengthens our argument that MUL1 suppresses PINK1 mutant phenotypes through reduction of Mfn levels.

*2) Figure 4M, O, S, U: does overexpression of parkin or MUL1 proteins reduce the level of overexpression of mitofusin protein*?

Yes. The data are shown in Figure 3.

*3)*
Figure 5*: why don't the* PINK1 *and* parkin *mutants shown here look much better (i.e., much less defective) than those shown in multiple panels of*
Figures 1, 2 and 4?

We used anti-ATPase antibody instead of mitoGFP to visualize mitochondrial morphology in Figure 5., which allows better visualization of the enhancement phenotypes seen in double mutants. To clarify this point, we now added the following sentence in the figure legend: “Instead of using mitoGFP, we utilized anti-ATPase antibodies, which allows better visualization of the enhancement phenotypes seen with double mutants.”

*4)*
Figure 8*: what effect do these various manipulations have on mitofusin levels in cortical neurons*?

Knockdown of MUL1 results in an increase in Mfn levels in mouse cortical neurons. In the Revision, we added Figure 8—figure supplement 2 to demonstrate this

Reviewer 2:

*This manuscript concludes that* Mul1*, a mitochondrial E3 ligase, ubiquitinates and causes proteosomal degradation of mitofusin in* Drosophil*a and mammalian cells. Furthermore,* Mul1 *loss exacerbates the phenotype of* Parkin *loss. The loss of* Mul1 *is clearly shown in*
Figure 5
*to make the loss of* Parkin/Pink1 *mutant flies worse, which indicates they work by different mechanisms. However, this is not surprising – many mutations in flies will cause stress by different paths and exacerbate one another's phenotype. Therefore, the key issue is whether or not both* Mul1 *and Parkin function by inducing the ubiquitination and elimination of mitofusin.* Mul1 *decreases the expression level of Mfn1 but this may be indirect and cell free experiments could conclusively show* Mul1 *ubiquitinates Mfn. However, cell free experiments seems to be beyond the scope of this manuscript. Why would increased Mfn level lead to the mitochondrial and cellular defects? Is this through decreased mitophagy? If not mitophagy what? MUL1 is not likely expressed simply to statically reduce the steady state level of Mfn. How MUL1 is regulated and why and when it ubiquitinates MFn (if it is direct) remains unexplored. Although there are clearly mysteries here, this is a very thorough genetic study with intriguing conclusions in need of only minor corrections*.

These are excellent questions for future studies.

Reviewer 3:

*[…] The subtle phenotypes associated with* MUL1 *mutants suggest a compensatory role of this pathway. The authors should highlight the importance of* MUL1 *in physiological conditions. Can the authors document, either by loss of one copy of* Pink1 *or* Parkin *or by stressing mitochondria, a stronger phenotype in* MUL1 *mutants. They should test life span or fertility in* MUL1 *mutants under normal and stressed conditions to provide a comparison with* pink/parkin *mutant phenotypes*.

These are good experiments and we plan to carry out these studies in the near future.

[Editors' note: further clarifications were requested prior to acceptance, as described below.]

*The authors have addressed most of the comments that were raised in the original review, with one notable exception. The authors have not shown that there is a direct physical interaction between the endogenous Mul1 and Mfn proteins. The authors state that this is not possible, because there is no antibody against* Drosophila *Mul1 and the antibody against mammalian Mul1 does not work for IP. However, there are antibodies available against mammalian Mfn that have been reported to work for immunoprecipitation. This experiment should be done, as originally requested, or the authors should present a strong argument for why it cannot be done*.

We agree with the editors that it is important to demonstrate a direct physical interaction between the endogenous MUL1 and Mfn. We have made repeated attempts to carry out this experiment in mammalian cells, using both anti-MUL1 and anti-Mitofusin antibodies. However, we have not been able to detect consistent binding between the endogenous proteins. It is possible that the interaction is too weak or transient, and that it therefore evades detection in our system, even though we can detect interactions between the two proteins when both are overexpressed.

It may be useful to provide a point of comparison with another well-established Mfn-binding protein, Parkin. None of the papers detailing interactions between Parkin and Mfn, and Parkin-dependent ubiquitination of Mfn, have shown these interactions using Co- IP between endogenous proteins (8; 22; 25; 54; 59; 70). While this state of affairs is not ideal, and it is not our preferred form of rebuttal, it does highlight what may be a general difficulty in detecting interactions between endogenous Mfn and other proteins. This has not prevented people in the field, including us, from concluding that Parkin binds to Mfn.

Finally, we interpret our data cautiously in the Discussion: “These observations suggest that MUL1 may directly ubiquitinate Mfn, leading to its degradation. Alternatively, MUL1 may act via an intermediary that promotes Mfn ubiquitination and degradation”.